# State-specific morphological deformations of the lipid bilayer explain mechanosensitive gating of MscS ion channels

Yein Christina Park[1], Bharat Reddy[2], Navid Bavi[2], Eduardo Perozo[2]*, José D Faraldo-Gómez[1]*

[1]Theoretical Molecular Biophysics Laboratory, National Heart, Lung and Blood Institute, National Institutes of Health, Bethesda, United States; [2]Department of Biochemistry and Molecular Biology, University of Chicago, Chicago, United States

**Abstract** The force-from-lipids hypothesis of cellular mechanosensation posits that membrane channels open and close in response to changes in the physical state of the lipid bilayer, induced for example by lateral tension. Here, we investigate the molecular basis for this transduction mechanism by studying the mechanosensitive ion channel MscS from Escherichia coli and its eukaryotic homolog MSL1 from Arabidopsis thaliana. First, we use single-particle cryo-electron microscopy to determine the structure of a novel open conformation of wild-type MscS, stabilized in a thinned lipid nanodisc. Compared with the closed state, the structure shows a reconfiguration of helices TM1, TM2, and TM3a, and widening of the central pore. Based on these structures, we examined how the morphology of the membrane is altered upon gating, using molecular dynamics simulations. The simulations reveal that closed-state MscS causes drastic protrusions in the inner leaflet of the lipid bilayer, both in the absence and presence of lateral tension, and for different lipid compositions. These deformations arise to provide adequate solvation to hydrophobic crevices under the TM1-TM2 hairpin, and clearly reflect a high-energy conformation for the membrane, particularly under tension. Strikingly, these protrusions are largely eradicated upon channel opening. An analogous computational study of open and closed MSL1 recapitulates these findings. The gating equilibrium of MscS channels thus appears to be dictated by opposing conformational preferences, namely those of the lipid membrane and of the protein structure. We propose a membrane deformation model of mechanosensation, which posits that tension shifts the gating equilibrium towards the conductive state not because it alters the mode in which channel and lipids interact, but because it increases the energetic cost of the morphological perturbations in the membrane required by the closed state.

*For correspondence:
eduardo.perozo@uchicago.edu
(EP);
jose.faraldo@nih.gov (JDF-G)

## Editor's evaluation

The manuscript reports a new structure of the small conductance mechanosensitive channel MscS from *E. coli* in the open state, together with coarse-grained and atomistic molecular dynamics simulations of MscS and the related channel MSL1 of plant mitochondria in closed and open states. The important finding is that the surrounding lipid bilayer is severely distorted in the closed state only, with the protein inducing high curvature in the inner leaflet due to the membrane protruding into the cytoplasm. The authors argue convincingly that the role of membrane tension is to increase the energy of the protein-membrane system in this closed state compared to the relatively flat-membrane open state, in contrast to the previous proposal that tension-induced gating is driven by expansion of the in-plane area of the protein. The finding may be relevant for

the understanding of ion channel mechano-sensation more generally, including of the PIEZO1 channel.

## Introduction

Sensory perception is a defining characteristic of life. In a constantly changing environment, cells must be able to detect a variety of stimuli, ranging from electric fields and chemical gradients to temperature and pressure. To respond to these inputs, cells have evolved a wide array of specialized membrane proteins; among these are mechanosensitive ion channels, which are activated by mechanical forces exerted on the cell. Broadly speaking, mechanosensitive channels fall into two classes: those that respond to changes in membrane tension and those that transduce other structural perturbations. Examples in the latter class include NOMPC channels, which open upon compression of their molecular structure against the microtubule network to which they are tethered (*Zhang et al., 2015*). PIEZO, by contrast, belongs to the class of channels directly sensitive to lateral membrane tension (*Katta et al., 2015*; *Cox et al., 2018*; *Guo and MacKinnon, 2017*), through a 'force-from-lipids' mechanism (*Kung, 2005*; *Anishkin et al., 2014*); this class also includes MscL and MscS, two extensively characterized channels whose physiological role is to relieve osmotic stresses so as to prevent cell rupture (*Katta et al., 2015*; *Cox et al., 2018*; *Guo and MacKinnon, 2017*; *Kung et al., 2010*; *Haswell et al., 2011*).

The force-from-lipids hypothesis, while firmly established by functional assays, remains to be rationalized at the molecular and physical levels. One challenge is that each mechanosensitive channel family has a distinct prototypical molecular structure (*Katta et al., 2015*; *Cox et al., 2018*; *Kung et al., 2010*; *Wilson et al., 2013*; *Flegler et al., 2020*). This divergence likely reflects that cells need to detect a variety of mechanical stimuli with different degrees of sensitivity (*Guo and MacKinnon, 2017*). It is thus conceivable that evolution has led to different realizations of the force-from-lipids principle as well. Even so, it is worth noting that channels within the same family, and hence a shared molecular mechanism, operate in vastly different membrane contexts. MscS channels, for example, have been identified in bacteria, archaea, fungi, and plants (*Wilson et al., 2013*; *Pivetti et al., 2003*).

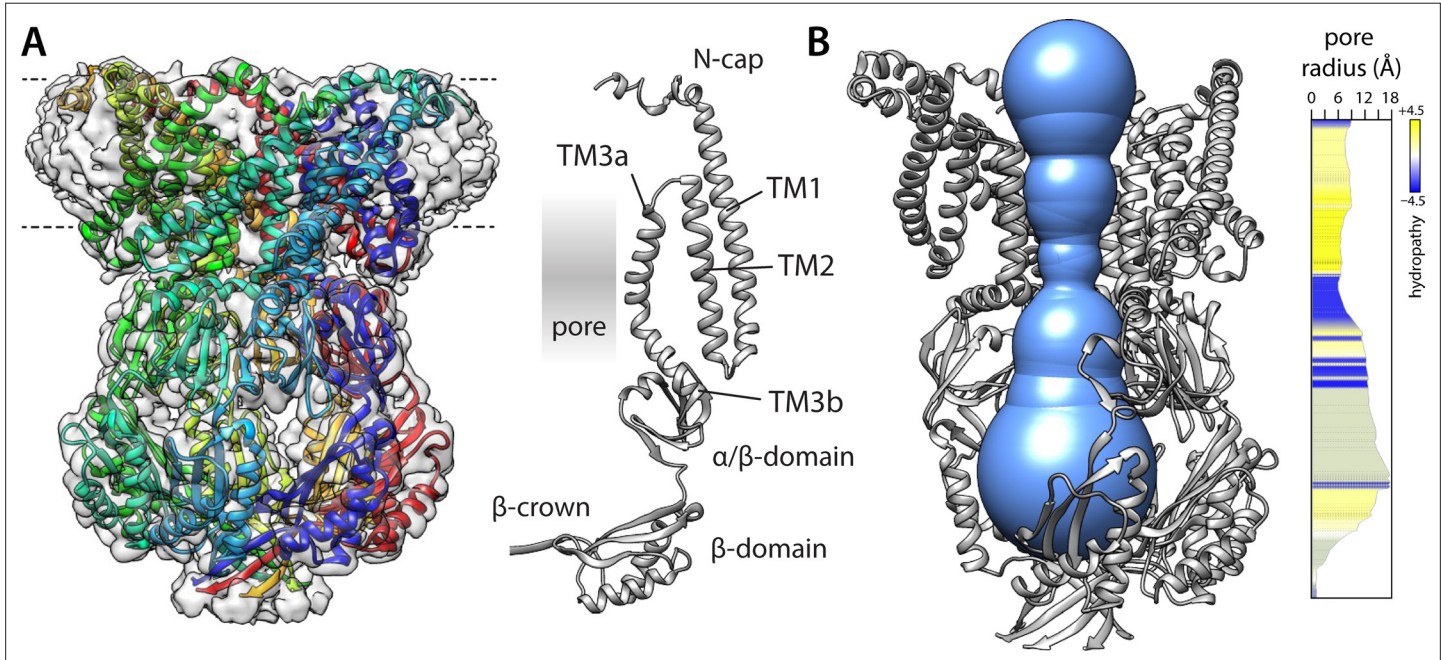

**Figure 1.** Structure of a putatively open conformation of wild-type MscS in a lipid nanodisc. (**A**) Left, structure of the MscS heptamer (in cartoon representation) fitted on the 3.1 Å resolution cryo-electron microscopy (EM) map (transparent surface). Each protomer is shown in a different color. The approximate position of the membrane is indicated with dashed lines. Right, cartoon representation of an individual protomer, in relation to the central pore. (**B**) Cut-away of the structure (minus two protomers) showing the shape of the ion permeation pathway as determined by MOLE*online* (*Pravda et al., 2018*). The rightmost plot quantifies the radius along the central symmetry axis as well as the hydrophobicity of the pore amino-acid make up.

This ubiquity across kingdoms of life raises intriguing questions: what single mechanism of tension sensing could accommodate these very different lipid environments? And might that mechanism be transferable not only across species but also among entirely different families of mechanosensitive channels?

Here, we focus on two channels of the MscS family that have been structurally characterized in multiple functional states, namely the *Escherichia coli* mechanosensitive channel of small conductance, EcMscS (or MscS) and its homolog in *Arabidopsis thaliana*, AtMSL1 (or MSL1). MscS resides in the inner membrane of *E. coli* and greatly increases its open-state probability under moderate membrane tensions (5.5 mN/m in liposome patches [*Kung et al., 2010*; *Haswell et al., 2011*]), thereby permitting passive diffusion of both ions and water (*Cox et al., 2018*). All available MscS structures, obtained through either X-ray crystallography or cryo-electron microscopy (EM; *Bass et al., 2002*; *Wang et al., 2008*; *Pliotas et al., 2015*; *Flegler et al., 2021*; *Zhang et al., 2021*; *Reddy et al., 2019*), show that this channel is a homoheptamer, featuring a transmembrane domain and a cytosolic domain. In the transmembrane domain, so-called helix TM3a from each protomer lines a central pore that serves as the ion conduction pathway, flanked by helices TM1 and TM2, which face the lipid bilayer (*Figure 1*). It is through changes in the tertiary and quaternary structure of this domain that MscS opens and closes. Specifically, in each protomer, the TM1–TM2 hairpin changes tilt relative to both TM3a and to the membrane plane, and in turn, results in widening or narrowing of the central pore. The magnitude of these changes has not been entirely clear, though, as the conducting state of the channel has been more difficult to capture than the closed form. Putatively open states have thus far resulted from either stabilizing point mutations (*Wang et al., 2008*; *Pliotas et al., 2015*; *Zhang et al., 2021*) or from solubilization in extremely thin lipid bilayers (*Zhang et al., 2021*) or in specific detergents (*Flegler et al., 2021*), leading to slight differences in the observed structures of the transmembrane domain. In contrast, the cytosolic domain and the elements that connect it with the pore (i.e. TM3b) are largely unchanged during channel gating; this domain thus appears to function as a sieve to prevent leakage of metabolically important small molecules (*Kung et al., 2010*), while enhancing the stability of the oligomeric complex, whether in the closed or open state.

Structures of closed and open states of MSL1 show a great degree of similarity with MscS, both in their overall architecture as well as in the conformational changes involved in gating. MSL1 is localized in the inner membranes of the *A. thaliana* mitochondria, where it is believed to contribute to dissipating the membrane potential under certain physiological conditions (*Deng et al., 2020*). When expressed in *E. coli*, MSL1 can protect the bacteria from hypo-osmotic stress, although much less effectively than MscS (*Li et al., 2020*). MSL1 is predicted to feature two additional N-terminal helices, for a total of five (TM[−1] to TM3, following MscS nomenclature), but these additional elements have not been resolved in existing structures (*Deng et al., 2020*; *Li et al., 2020*). Its pore-forming unit is however equivalent to that of MscS.

While the structural mechanism associated with MscS channel gating appears to be largely delineated, how this mechanism is influenced by membrane stretching has been an elusive question. Lateral tension is known to alter several bulk properties of the lipid bilayer, such as thickness or spontaneous curvature (*Haswell et al., 2011*; *Marrink and Mark, 2001*); however, it is important to recognize that any model of mechanosensation predicated on major modifications of membrane properties is likely implausible, as it presupposes that these changes are not deleterious to other proteins embedded in or associated with the lipid bilayer. Moreover, it is unclear whether the moderate tensions that activate MscS can indeed effect changes in these properties of sufficient magnitude to explain the sensitivity of these channels, or importantly, why these putative changes would specifically favor the open state. Indeed, when attempting to formulate a theory of mechanosensation, the challenge lies not only in identifying molecular features that are specific to the channel in question and that vary with its functional state; it is also crucial that these distinctive features be differentially impacted by lateral tension, in a manner that favors the open state or disfavors the closed state.

In this study, we present structural and computational data that lead to a model of mechanosensation that satisfies these criteria. Specifically, we observe that the morphology of the lipid bilayer in the proximity of the protein is strongly deformed for the closed but not the open state and propose that tension shifts the gating equilibrium in favor of the open state not because it fundamentally alters the nature of the interactions between protein and lipids but because it increases the energetic cost of the membrane deformations that sustain the closed-state structure. This model does not presuppose

the existence of long-lasting protein-lipid interactions that are somehow weakened under tension. Instead, our view recognizes that the lipid bilayer is a structure with defined morphological energetics and, at the same time, a highly concentrated liquid solvent in constant motion at the molecular level.

## Results

### Structure of an open conformation of MscS in a PC14:1 lipid bilayer

To determine an open-state structure of wild-type (wt) MscS in conditions that resemble a membrane, we purified the channel and reconstituted it into nanodiscs (NDs) consisting of myristoyl-phosphatidylcholine PC14:1 lipids and MSP1 E3D1 scaffold proteins, which we then imaged using single-particle cryo-EM. (Alternative preparations of the wt channel using PC14:0 lipids or of the A106V mutant in dioleoyl-phosphatidylcholine PC18:1 NDs failed to reveal an open state and instead resulted in a conformation indistinguishable from that of the known closed state, not shown.) The resulting cryo-EM structure of wt-MscS in PC14:1 is shown in *Figure 1A* (*Supplementary file 1*). The resolution of the map is 3.1 Å; density signals can be discerned and traced for most of the polypeptide chain, except for residues 1–15 at the N-terminus, for which sidechain assignments required using flexible fitting. The structure recapitulates known features of the MscS heptameric architecture; that is, a large intracellular domain, which features numerous contacts among protomers and appears to facilitate oligomerization, and a much more loosely packed transmembrane domain, which responds to the physical state of the membrane and which consists of helices TM1, TM2, and TM3 in each protomer. As observed in previous structures of MscS (*Bass et al., 2002*; *Wang et al., 2008*; *Pliotas et al., 2015*; *Flegler et al., 2021*; *Zhang et al., 2021*; *Reddy et al., 2019*), TM3 is not a continuous helix but breaks into two distinct segments, approximately halfway. The N-terminal fragment, TM3a, flanks much of the transmembrane portion of the ion permeation pathway (*Figure 1B*), while TM1 and TM2 are positioned peripherally toward the lipid bilayer. The C-terminal fragment of TM3, TM3b, serves as the linker between the transmembrane and intracellular domains. It is at the junction between TM3a and TM3b where the ion permeation pathway is narrowest in this structure; however, at this point, the pore diameter is ~13 Å, which is sufficient to permit flow of hydrated cations. The cryo-EM map reveals no evidence of any obstructions, such as lipids, in this seemingly open pore.

*Figure 2A* compares the structure of this putatively open state of wt-MscS obtained in PC14:1 lipid NDs and that of the closed state we previously determined in PC18:1 (*Reddy et al., 2019*). No significant differences exist in the intracellular domain or its arrangement relative to the TM3b linker; the internal structure of the TM1–TM2 hairpin is also largely invariant, although TM1 is elongated through a reconfiguration of its N-terminal cap, which in the closed state lies parallel to the membrane surface (*Reddy et al., 2019*). By contrast, there is a drastic reorientation of the TM1–TM2 hairpin relative to TM3a as well as the membrane plane. Specifically, the TM1–TM2 hairpin tilts approximately by 40° toward the bilayer midplane upon channel opening; in turn, this reorientation permits TM3a to alter its angle relative to TM3b and to retract away from the pore axis, thereby widening the ion permeation pathway. This drastic change in the TM1–TM2 hairpin, which is symmetrically replicated in all protomers, translates into a marked reduction in the hydrophobic width of the channel (i.e. the length of the transmembrane span along the bilayer perpendicular). This reduction is likely one of the factors that explain why the open conformation is favored upon reconstitution in PC14:1 NDs, which are expected to be significantly thinner than the PC18:1 NDs that stabilize the closed state (*Reddy et al., 2019*). Indeed, this thinning is clearly discernable when the cryo-EM maps of closed and open states are compared (*Figure 2B*). The open-state map, however, seems to reflect a greater heterogeneity in the protein conformation and does not permit a conclusive assignment of individual lipid molecules to specific sites on the protein surface. Nonetheless, a series of densities atop the C-terminus of TM2, filling in small fenestrations formed on the extracellular face of the channel, appear to replicate some of the lipid-interaction sites we had previously detected in the closed state, which we termed 'hook' lipids (*Reddy et al., 2019*; *Figure 2—figure supplement 1*).

Lastly, *Figure 3* compares the structure of wt-MscS described above with two others reported previously and proposed to also capture open or partially open conformations of the channel: that of a mutant (A106V), crystallized in detergent micelles and resolved through X-ray diffraction (*Wang et al., 2008*) and that of wt-MscS reconstituted in PC10:0 lipid NDs, imaged with single-particle cryo-EM (*Zhang et al., 2021*). This comparison makes it clear that the extent of the conformational changes

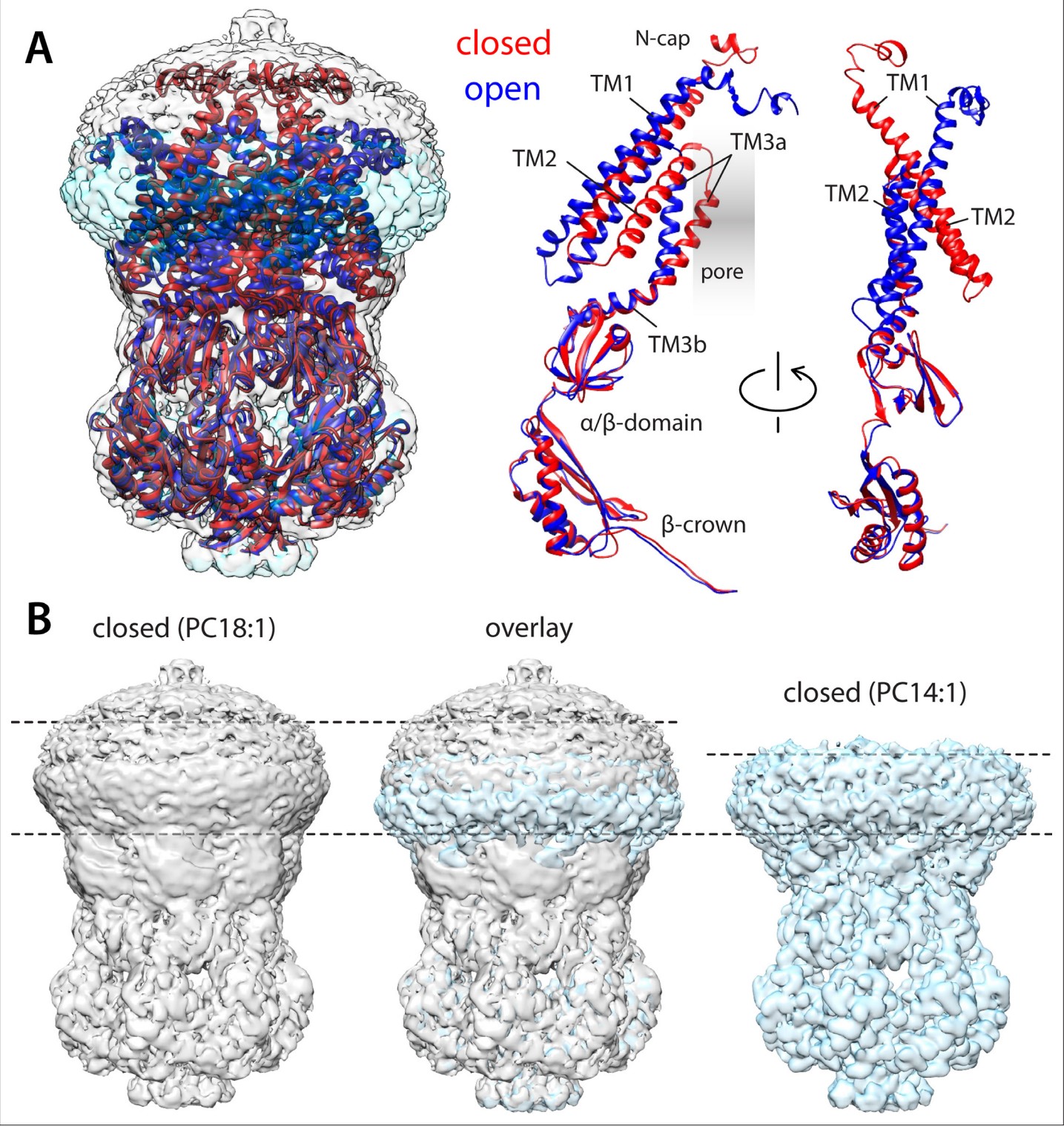

**Figure 2.** Comparison of closed and open structures of MscS in lipid nanodiscs. (**A**) Left, the open structure of MscS in PC14:1 lipid nanodiscs (blue cartoons) is superimposed onto that of the closed state (red), previously determined in PC18:1 nanodiscs (PDB ID 6PWN, and EMD-20508). The corresponding cryo-electron microscopy (EM) density maps (transparent cyan and gray surfaces, respectively) are shown as well. Right, conformational change in each of the protomers. (**B**) Side-by-side comparison of the cryo-EM density maps obtained in PC18:1 (gray) and PC14:1 (cyan), alongside their overlap. The comparison highlights the reduction in the width of the transmembrane span of the channel upon opening, seemingly matched by thinning of the lipid nanodiscs, by approximately 7 Å.

The online version of this article includes the following figure supplement(s) for figure 2:

*Figure 2 continued on next page*

*Figure 2 continued*

**Figure supplement 1.** Cryo-electron microscopy (EM) map and structural model of open-state MscS in PC14:1 lipid nanodiscs, highlighting putative sites of lipid interaction atop the C-terminus of TM2, involving residue R88.

observed in our structure is significantly amplified in those existing structures. For example, TM1 tilts by up to 55°, and TM2 tilts by about 45° (*Figure 3A*). However, these differences have a minor impact on the dimensions of the pore, whose diameter is, at most, only 2 Å wider (*Figure 3B*). Admittedly, none of the three structures capture the channel embedded in a native membrane (arguably, though, a PC14:1 ND is a more realistic mimic than a detergent micelle or a PC10:0 ND). Since the conformational changes observed in the PC14:1 structure appear sufficient to open up the ion permeation pathway, we posit that the structure of wt-MscS reported here is, at minimum, similarly likely to represent the conductive form of the channel in physiological conditions.

## Closed-state MscS induces drastic deformations of the lipid bilayer

To begin to investigate how the physical state of the lipid bilayer might influence the conformational equilibrium between conductive and non-conductive forms of MscS, we first examined the membrane morphology associated with the closed state of the channel, using the structure we previously resolved in PC18:1 lipid NDs (*Reddy et al., 2019*). To that end, we carried out a series of molecular dynamics (MD) simulations, based on both coarse-grained (CG) and all-atom (AA) representations, evaluating multiple lipid bilayer compositions and membrane dimensions (*Table 1*). All CG trajectories lasted for 20 μs each and were initiated with a flat membrane; the simulations were calculated independently of any experiments, and no prior assumptions were made in regard to the configuration of the protein-lipid interface (see Materials and methods for further details). The AA trajectories were initiated from a representative configuration of a CG trajectory obtained under the same conditions and lasted 10 μs each. Invariably, all these simulations showed that stabilization of the closed conformation of MscS demands drastic deformations in the morphology of the lipid bilayer, in particular in the inner leaflet. *Figure 4* depicts these deformations using 3D density maps derived from the simulated MD trajectories for several of the different conditions explored. These density maps clearly reveal extensive protrusions in the inner leaflet of the membrane, which develop to provide adequate lipid solvation

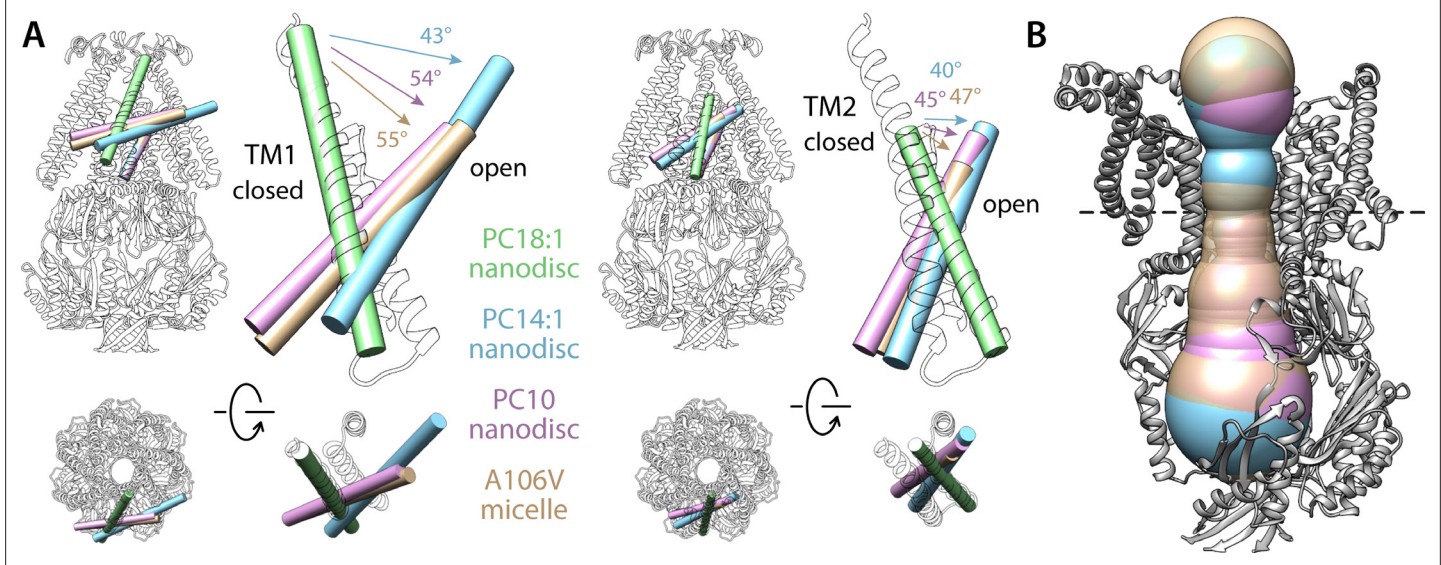

**Figure 3.** Comparison of alternate conformations of open MscS obtained in different experimental conditions. (**A**) Overlay comparing helices TM1 and TM2 in the closed state (green tube and white cartoons) and in the putatively open states obtained in PC14:1 nanodiscs (this work, cyan), in PC10:0 nanodiscs (PDB 6VYL, pink) and in detergent micelles (PDB ID 2VV5, orange). Note the structures in lipid nanodisc were determined with single-particle cryo-electron microscopy (EM) for the wild-type protein, while the A106V structure in detergent was determined with X-ray crystallography. All structures are superimposed based on TM3. (**B**) Comparison of the dimensions of the permeation pore in the three structures of the putatively open state. The black-dashed line indicates the position of L105, which is approximately where the pore is narrowest; the radius here is 6.3 Å for the PC14:1 structure, 6.7 Å for the PC10:0 structure, and 7.5 Å for the mutant in detergent.

**Table 1.** Molecular simulation systems evaluated in this study of McsS and MSL1 mechanosensation.

| | MscS coarse grained* | | | | | | | MscS all atom† | | MSL1 coarse grained* | |
|---|---|---|---|---|---|---|---|---|---|---|---|
| | Closed WT POPC | Closed WT POPC §‡ | Closed WT DMPC | Closed WT PC:PG | Open WT POPC | Open A106V POPC | Open D67R1 POPC | Closed WT POPC | Closed WT DMPC | Closed WT POPC | Open WT POPC |
| Protein | 1 | 1 | 1 | 1 | 1 | 1 | 1 | 1 | 1 | 1 | 1 |
| POPC | 2057 | 755 | 0 | 1651 | 686 | 737 | 740 | 755 | 0 | 2001 | 1916 |
| POPG | 0 | 0 | 0 | 407 | 0 | 0 | 0 | 0 | 0 | 0 | 0 |
| DMPC | 0 | 0 | 768 | 0 | 0 | 0 | 0 | 0 | 768 | 0 | 0 |
| Na$^+$ | 855 | 328 | 340 | 1265 | 328 | 328 | 328 | 314 | 326 | 855 | 855 |
| Cl$^-$ | 862 | 335 | 347 | 865 | 335 | 356 | 349 | 335 | 347 | 890 | 883 |
| Water ‡ | 75,938 | 28,123 | 29,000 | 75,501 | 28,200 | 28,372 | 28,299 | 115,170 | 118,774 | 76,076 | 76,549 |
| Total atoms or particles | 106,364 | 41,871 | 41,392 | 106,352 | 41,120 | 41,603 | 41,524 | 476,183 | 476,473 | 106,012 | 105,248 |
| System size (nm) | 26.5×26.5×18.7 | 16.9×16.9×17.9 | 16.3×16.3×19.1 | 26.3×26.3×19.0 | 16.9×16.9×17.9 | 16.9×16.9×17.9 | 16.9×16.9×17.9 | 15.9×15.9×19.9 | 15.4×15.4×21.1 | 26.5×26.5×18.7 | 26.5×26.5×18.7 |
| Time (µs) | 20 | 20 | 20 | 20 | 20 | 20 | 20 | 10 | 8 | 80 | 20 |

POPC: 1-palmitoyl-2-oleoyl-sn-glycero-3-phosphocholine; WT: wild type; DMPC:1-2-dimyristoleoyl-sn-glycero-3-phosphocholine; POPG:1-2-palmitoyl-2-oleoylglycero-3-phosphoglycerol.

*MARTINI 2.2 forcefield.

†CHARMM36m forcefield.

‡One CG water particle is equivalent to four AA water molecules.

§Five additional 10 µs simulations were carried out for this system under applied lateral tensions of 0, 0.5, 2.5, 5.0, and 10 mN/m.

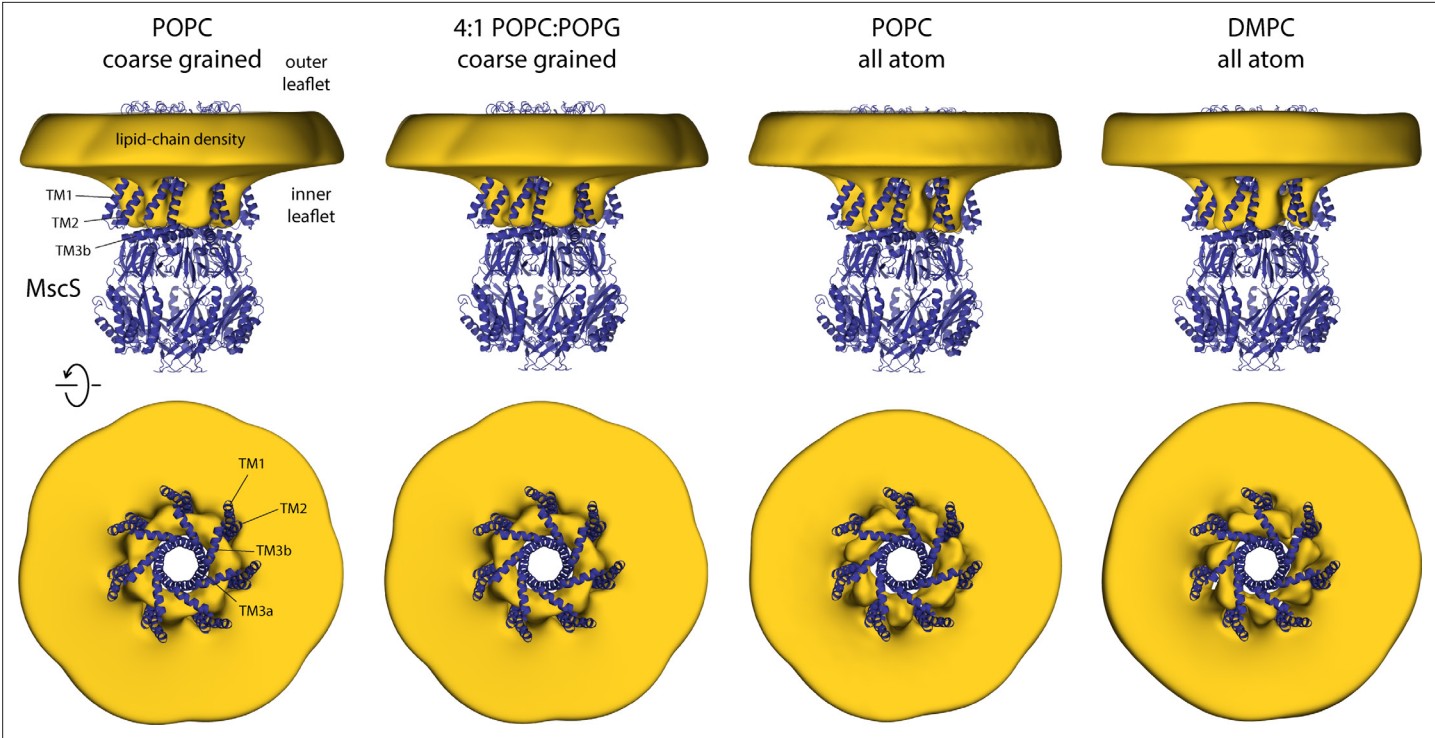

**Figure 4.** Closed-state MscS induces drastic perturbations of the lipid bilayer. The figure summarizes the results from multiple simulations of the closed structure of MscS in different membrane compositions and using different forcefield representations (*Table 1*). The cryo-electron microscopy (EM) structure of MscS (blue cartoons) is overlaid with calculated 3D density distributions mapping the morphology of the alkyl chain bilayer in each of the molecular dynamics (MD) trajectories (gold volume), up to 50 Å from the protein surface. Protein and density maps are viewed along the membrane plane (top row) and along the pore axis, from the cytosolic side (bottom row); the latter includes only the transmembrane domain of the channel, for clarity. The calculated density maps derive from 20 μs of trajectory data for each of the coarse-grained systems and at least 8 μs of trajectory data for the all-atom systems.

The online version of this article includes the following figure supplement(s) for figure 4:

**Figure supplement 1.** Lipid solvation of hydrophobic cavities outside the membrane drives the formation of inner-leaflet protrusions in closed-state MscS.

**Figure supplement 2.** Relaxation of the closed-state MscS structure in all-atom simulations.

to exposed hydrophobic surfaces in the protein in this particular state (*Figure 4—figure supplement 1*). These hydrophobic surfaces line crevices formed between the TM1–TM2 hairpin and TM3a–TM3b, well outside what might be predicted as the transmembrane span of the channel; lipid solvation of these cavities thus requires dramatic morphological changes in the bilayer. Logically, the simulations reveal seven nearly identical protrusions, matching the symmetry of the channel structure.

To discern the structure of these membrane perturbations more precisely, we analyzed the average molecular configuration of the lipids found in these regions as well as elsewhere in the bilayer. As shown in *Figure 5*, it is striking that to form these protrusions, the headgroup layer of the inner leaflet must not only project away from the membrane center but also bend sharply, thus permitting the underlying acyl chains to become strongly tilted, by as much as 60° relative to the membrane perpendicular (*Figure 5—figure supplement 1*). Importantly, examination of the lipid translational dynamics makes it clear that the molecules in these protrusions are in constant exchange with the rest of the bilayer (*Figure 6*). As expected, the turnover of the lipids found in the protrusions is slower than elsewhere, but only by an order of magnitude. This observation is comparable for the CG and AA systems; naturally, lipid diffusion is artificially accelerated in the CG representation, but it seems fair to conclude from the AA trajectories that the lipid content of the inner leaflet protrusions would be renewed multiple times within tens of microseconds. Thus, in the context of the membrane and at physiological temperature, it would hardly be accurate to characterize these lipid molecules as 'bound', as one would a conventional agonist or antagonist of a ligand-gated channel, for example.

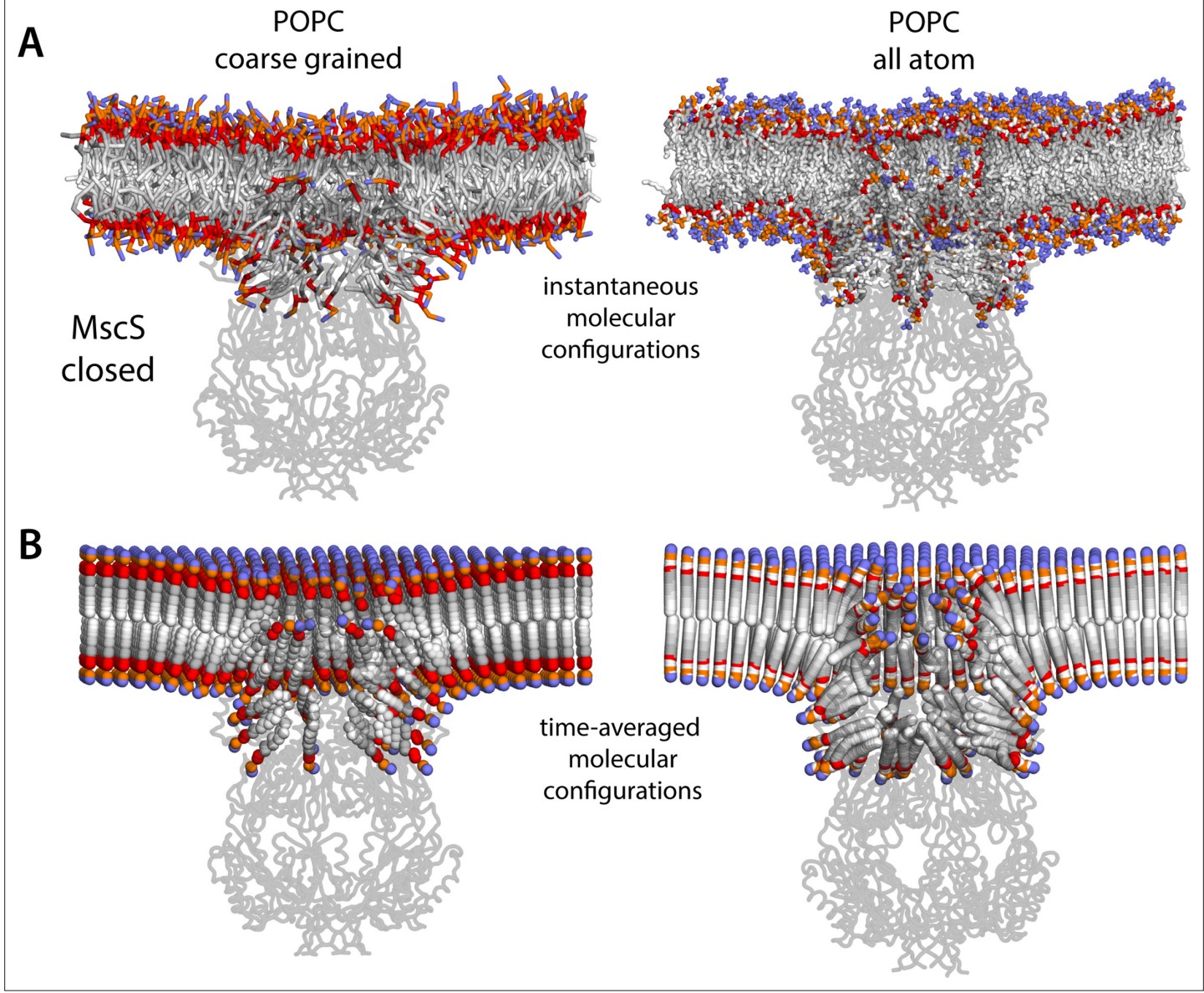

**Figure 5.** Molecular structure of the membrane perturbations induced by MscS in the closed state. (**A**) Instantaneous configurations of the lipid bilayer in single snapshots of two of the molecular dynamics (MD) trajectories calculated for closed-state MscS in POPC (1-palmitoyl-2-oleoyl-sn-glycero-3-phosphocholine), shown in cross section to reveal the structure of the membrane perturbations induced by the protein. Choline groups are shown in purple, phosphate in orange, ester linkages in red, and alkyl chains in shades of gray. The channel structure is overlaid, shown in cartoons. (**B**) Time-averages of the instantaneous lipid configurations observed in the same two trajectories, mapped across the membrane plane. Averages are calculated for each lipid atom and shown as spheres, colored as in panel (**A**). Owing to the configurational and rotational dynamics of lipids, the resulting averages are non-physical structures, but they nevertheless capture the position and orientation adopted by lipid molecules in different regions of the membrane. The average configuration of the channel backbone is also shown, overlaid. Protein and lipid averages are derived from 20 μs of trajectory data for the coarse-grained system and 10 μs of trajectory data for the all-atom system. The figures in panel A show snapshots at t ≈ 7 μs of the coarse-grained simulation and at t ≈ 5 μs of the all-atom simulation.

The online version of this article includes the following figure supplement(s) for figure 5:

**Figure supplement 1.** Molecular structure of the membrane perturbations induced by MscS in the closed state.

Instead, what we observe in these simulations is a reorganization of the structure and dynamics of the lipid solvent to adequately solvate the topography and amino-acid make-up of the protein surface in this particular state.

Finally, it is worth noting that the channel structure also bends the outer leaflet of the membrane, although to a much smaller degree (*Figure 5*). The most significant perturbation in the outer leaflet

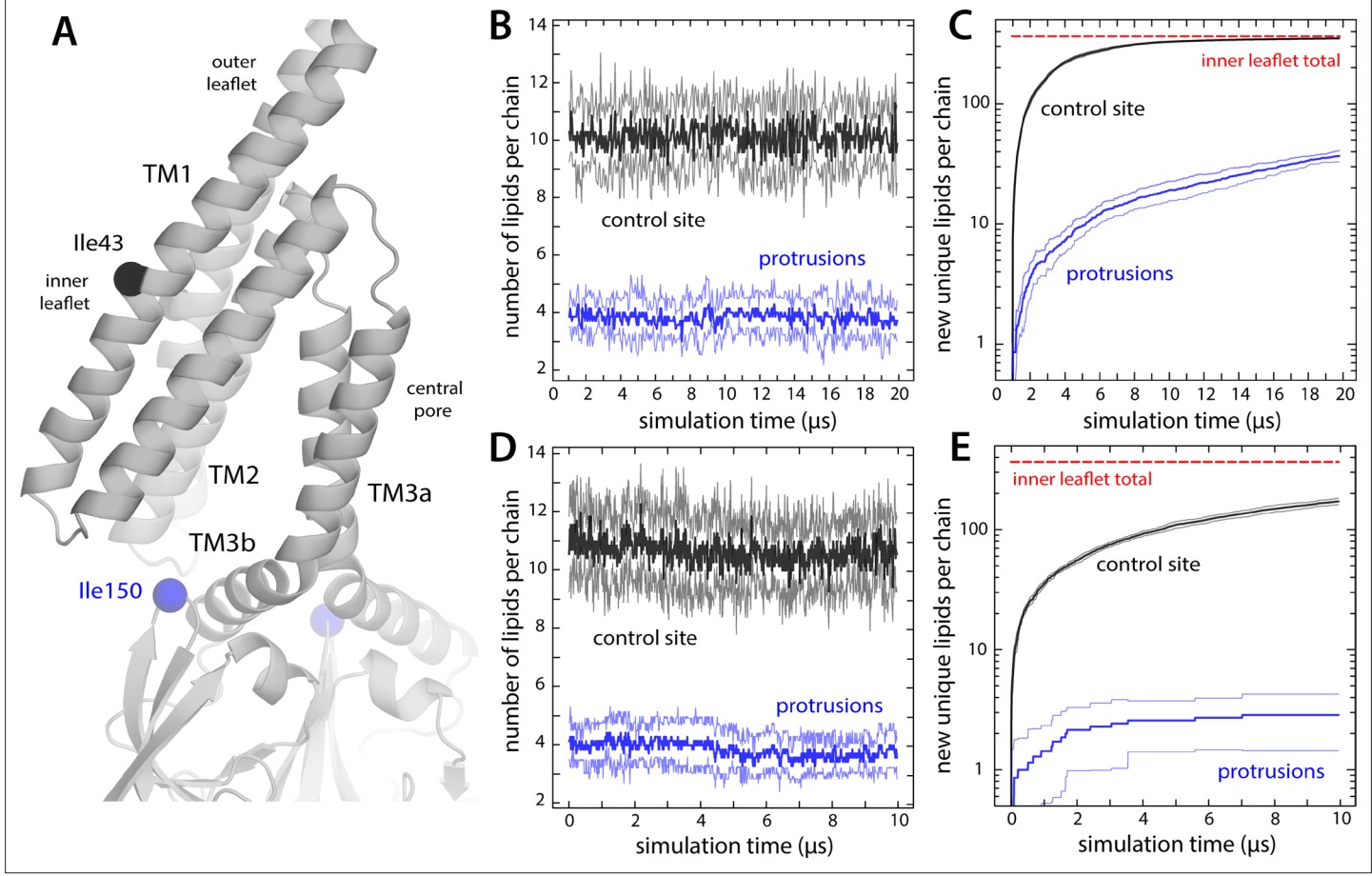

**Figure 6.** Lipids in inner-leaflet protrusions exchange with lipids in the bulk membrane. (**A**) Close-up of one of the hydrophobic cavities that drive the formation of membrane protrusions in closed-state MscS. To identify which lipids reside in these protrusions and for how long, we monitored their distance to the Cα atom of residue Ile150; for comparison, we also tracked their distance to residue Ile43, which faces the inner leaflet of the (locally unperturbed) membrane. (**B**) Number of lipid molecules found at any given time in the inner leaflet protrusions (i.e. within 20 Å of Ile150; blue lines), in comparison with those found in proximity to Ile43 (black), during coarse-grained simulations of closed-state MscS in POPC (1-palmitoyl-2-oleoyl-sn-glycero-3-phosphocholine). The plot shows the average value for the seven protrusions (thick lines) as well as the variance in the observed values (thin lines). (**C**) Time-course of the number of new, unique lipids contributing to the populations shown in panel (**B**). Lipids in the first snapshot were not considered newly found; hence, all curves start from a value of 0. For comparison, the total number of lipid molecules in the inner leaflet is indicated (red line). Note that after 20 μs of coarse-grained (CG) simulation, all lipids in the inner leaflet have at some point been in proximity to each of the seven Ile43 residues, while about 40 different lipids have been at some point in each of the seven membrane protrusions. (**D and E**) Same as (**B and C**), for the analogous 10 μs all-atom simulation of closed-state MscS in POPC.

is the sequestration of seven lipids (one per protomer), each of which fills a small fenestration in between adjacent TM1 helices, atop the C-terminus of TM2; the polar headgroups of these lipids project toward the aqueous interior of the pore and interact with R88. These molecules correspond exactly to density features in our cryo-EM map of the closed state (*Reddy et al., 2019*), and others reported subsequently (*Zhang et al., 2021*), which we referred to as 'hook lipids'. However, our simulations show that these lipid molecules originate from the outer leaflet of the membrane, not the inner leaflet, as previously theorized (*Zhang et al., 2021*).

## Lateral tension alone does not deplete lipids from membrane protrusions

In the literature on the gating mechanisms of channels that are sensitive to membrane stretching, it is often assumed that the nature of the interactions between these channels and the surrounding lipid molecules is fundamentally altered by the application of lateral tension; the term 'force-from-lipids' (which aptly draws a contrast with other forms of mechanotransduction, e.g. via the microtubule

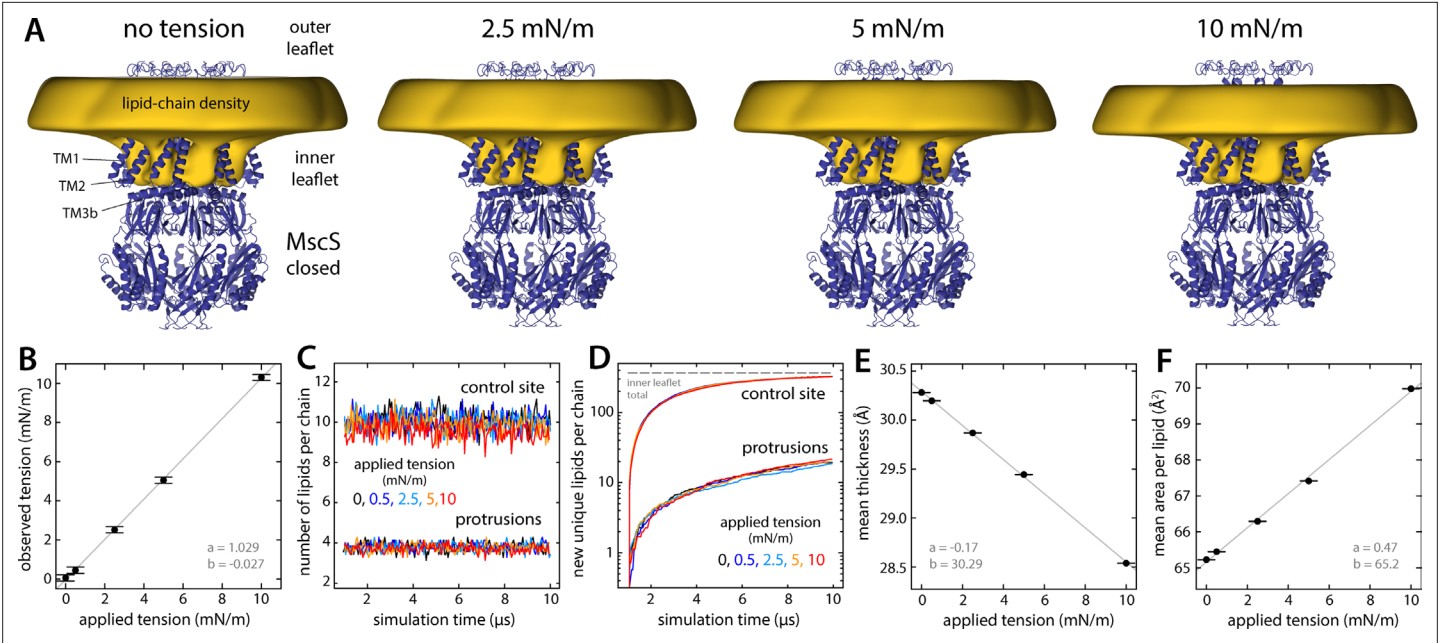

**Figure 7.** Persistence of lipid protrusions across increasing lateral tension conditions despite changes in global membrane properties. The figure summarizes the results from simulations of the closed structure of MscS under different membrane tensions. (**A**) The cryo-electron microscopy (EM) structure of MscS in the closed state (blue cartoons) is overlaid with a calculated 3D density distribution mapping the morphology of the alkyl chain double layer in the molecular dynamics (MD) trajectory (gold volume), up to 50 Å from the protein surface. Protein and density maps are shown as in **Figure 4**. (**B**) The lateral tension reported by the MD engine during the simulations (in 500 ps intervals, then averaged) is compared with the target tension value in each case (see Materials and methods). Error bars denote the SEM. A line of best fit is shown superimposed (gray), along with the slope *a* and intercept *b*. (**C**) Number of lipids in the inner-leaflet protrusions and near control sites, calculated as described in **Figure 6**, for each tension condition (average values for control sites: 10.13, 10.09, 10.01, 9.86, and 9.50 lipids; for protrusions: 3.82, 3.83, 3.80, 3.77, and 3.73 lipids; for applied tensions of 0, 0.5, 2.5, 5, and 10 mN/m, respectively). (**D**) Lipid exchange between the bulk and the protrusions/control sites, for each tension condition, evaluated as in **Figure 6**. (**E and F**) Changes in mean membrane thickness and mean area per lipid as the applied lateral tension increases. Bulk values were computed by averaging all local values (mapped on a discrete lattice) at least 20 Å away from the surface of the protein. Bilayer thickness was evaluated using the glycerol groups. Area-per-lipid values were averaged across leaflets. Error bars represent the SEM.

network) is unfortunate in that it reinforces this assumption. To evaluate this notion in the case of MscS, we carried out an additional series of simulations wherein the channel is held in the closed conformation (as is typical for CG simulations), while the lipid bilayer is stretched to varying degrees. Specifically, these simulations probed lateral tensions of 0.5, 2.5, 5, and 10 mN/m, which span an experimentally realistic range (MscS opens at ~5 mN/m [**Kung et al., 2010**; **Haswell et al., 2011**]); for comparison, we also carried out a control simulation targeting a zero lateral-tension value, using the same pressurization algorithm (which differs from that employed in the previous section – see Materials and methods).

The results of this analysis are summarized in **Figure 7**. It is apparent that the morphology of the lipid bilayer deformations induced by MscS in its closed state is largely invariant across the range of lateral tensions considered (**Figure 7A**). The simulated trajectories show no appreciable depletion of the lipid protrusions that fill the hydrophobic crevices under the TM1–TM2 hairpins (**Figure 7B**); applied lateral tension also does not accelerate the exchange of lipid molecules between the protrusions and the bulk membrane (**Figure 7C**). The topography and amino-acid make-up of the channel surface, therefore, appear to dictate the morphology and dynamics of the lipid bilayer in the vicinity of the protein in all conditions. This is not to say membrane stretching is inconsequential; indeed, the effect of tension is most clear when we examine bulk bilayer properties. For example, as tension increases, the membrane becomes thinner while the area per lipid expands, linearly in both cases (**Figure 7D**). These effects are subtle but occur across the entire membrane, including the protein-lipid interface, and make the bilayer stiffer, thereby increasing the energetic penalty of the deformations induced by the channel in its closed conformation. However, tension alone does not fundamentally alter the nature of the channel's interactions with the lipid bilayer; to appreciate why tension alters the

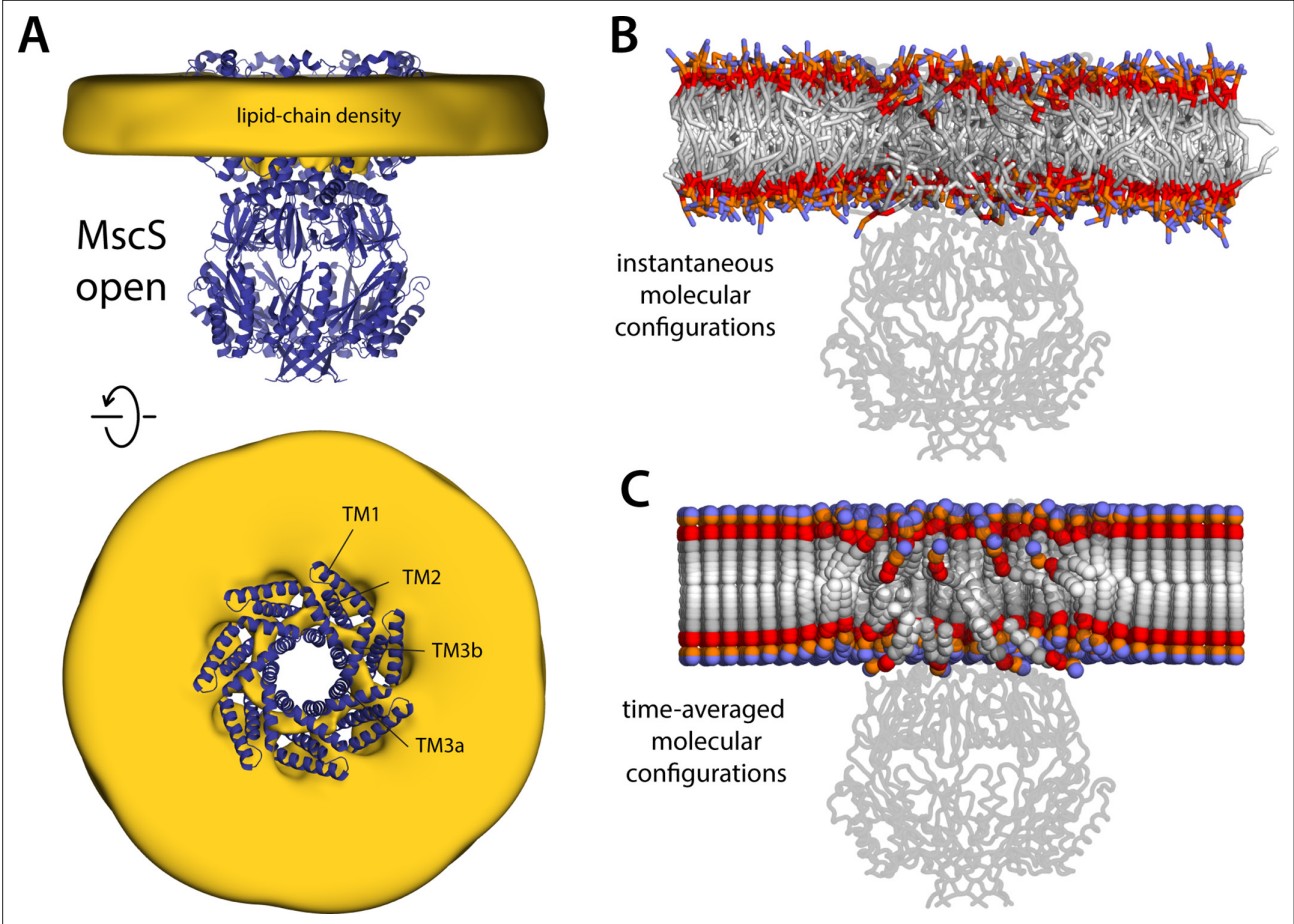

**Figure 8.** Membrane perturbations are largely eliminated upon MscS channel opening. The figure summarizes the results from a 20-μs simulation of open MscS in a POPC (1-palmitoyl-2-oleoyl-sn-glycero-3-phosphocholine) membrane, using a coarse-grained representation. (**A**) The cryo-electron microscopy (EM) structure of MscS in the open state (blue cartoons) is overlaid with a calculated 3D density distribution mapping the morphology of the alkyl chain double layer in the molecular dynamics (MD) trajectory (gold volume), up to 50 Å from the protein surface. Protein and density maps are shown as in *Figure 4*. (**B**) Instantaneous configuration of the lipid bilayer in a snapshot of the MD trajectory, shown in cross-section as in *Figure 5A*. (**C**) Time-averages of the instantaneous lipid configurations observed in the trajectory, mapped across the membrane plane and shown in cross-section. Averages were calculated and are represented as in *Figure 5B*.

The online version of this article includes the following figure supplement(s) for figure 8:

**Figure supplement 1.** Change in protein-lipid interfacial area during gating is much smaller than what could be inferred from change of in-plane cross-sectional area.

**Figure supplement 2.** Changes in membrane morphology upon gating of MscS.

gating equilibrium, one must examine the morphology of the membrane when the channel adopts the open state.

## MscS opening eliminates deformations in the lipid bilayer

To shift the gating equilibrium of an ion channel, a given stimulus such as membrane stretching must have a differential effect on the open and closed states. Following this reasoning, we next examined the morphology of the lipid bilayer that corresponds to the open state of MscS, using simulations of the new cryo-EM structure reported here. The central observation from this analysis is, strikingly, the near-complete eradication of the protrusions observed in the inner leaflet of the lipid bilayer for the closed state (*Figure 8*). This marked difference logically correlates with the structural changes observed in the channel: in the open state, the hydrophobic crevices formed between the TM1–TM2 hairpin and TM3a–TM3b are still filled with lipids, but the rotation of the TM1–TM2 hairpins aligns these crevices with the bulk membrane, and so these lipids do not protrude out (*Figure 8*), nor do

they become strongly tilted relative to the membrane perpendicular (*Figure 5—figure supplement 1*). By contrast, the perturbation of the outer leaflet is largely identical to that observed for the closed state; in particular, we observe 'hook' lipids sequestered between adjacent TM1 helices atop the C-terminus of TM2, as in the closed state. Thus, these 'hook' lipids appear to be structural, rather than a differential characteristic of one state or the other. It is also interesting to note that despite the expansion of the channel structure on the membrane plane , the total area of the protein-lipid interface, as quantified by the number of chemical groups in the channel structure that are in direct or close-range contact with lipid molecules, is only 3% greater for the open state, which is far smaller than what geometric idealizations would suggest (*Figure 8—figure supplement 1*). This discrepancy is due to the highly irregular morphology of the bilayer near the closed channel and illustrates the shortcomings of simplistic representations of the membrane, or the protein-lipid interface, in the context of mechanistic investigations.

For completion, we also carried out simulations of two other structures of MscS widely regarded to capture the open state of MscS, namely those of mutants A106V (*Wang et al., 2008*) and D67R1 (*Pliotas et al., 2015*). As discussed above, these structures were obtained in detergent and differ from our wt structure obtained in lipid NDs in the degree of tilt of the TM2–TM3 unit relative to the pore axis (*Figure 3*); another notable difference is that they do not resolve the first 24 N-terminal residues. These differences notwithstanding, the simulations carried out for these two structures reaffirm the results obtained for the open-state conformation reported here; that is, the hydrophobic crevices under TM1–TM2 units remain lipidated, but the sizable protrusions of the inner leaflet seen for the closed state are largely eliminated (*Figure 8—figure supplement 2*).

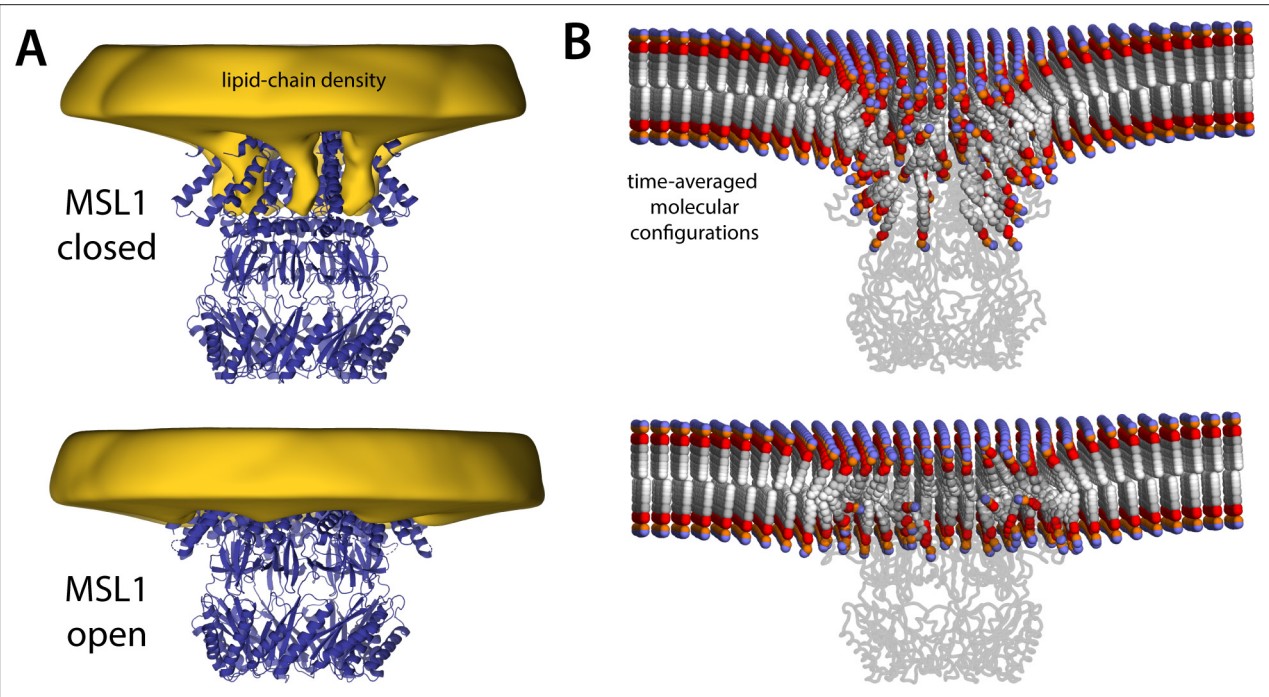

**Figure 9.** MSL1 gating causes morphological changes in membrane akin to those observed for MscS. The figure summarizes the results from simulations of closed and open states of MSL1 in a POPC (1-palmitoyl-2-oleoyl-sn-glycero-3-phosphocholine) membrane, using a coarse-grained representation (*Table 1*). (**A**) The cryo-electron microscopy (EM) structures (blue cartoons) are overlaid with calculated 3D density distributions mapping the morphology of the alkyl chain double layer in each of the molecular dynamics (MD) trajectories (gold volumes), up to 50 Å from the protein surface. Maps were calculated and are represented as in *Figure 4*. (**B**) Time-averages of the instantaneous lipid configurations observed in each of the two MD trajectories, mapped across the membrane plane and shown in cross-section. Averages were calculated and are represented as in *Figure 5B*. The datasets in panels (**A**) and (**B**) were symmetrized in accordance with the sevenfold internal symmetry of the channel.

## Mitochondrial MscS homolog also deforms the bilayer in the closed but not the open state

When considering the potential mechanistic significance of a given observation, it seems reasonable to evaluate its transferability across homologous systems in different species or, more broadly, across proteins featuring similar functional characteristics. This is not a trivial consideration in the case of mechanosensation, as the divergence between channels and species involves not only different protein sequences and entirely different folds but also membranes of entirely different lipid composition. For example, the inner membrane of *E. coli* is ~75% PE, 20% PG, and 5% CL (*Miyazaki et al., 1985*), whereas inner mitochondrial membranes are 40% PC, 30% PE, 15% CL, and other (*Schenkel and Bakovic, 2014*). To begin to ascertain whether the observations described above for MscS translate to other homologs in this family, we examined the lipid bilayer morphologies that correspond to the open and closed states of MSL1, the MscS-like channel from *A. thaliana*. As shown in *Figure 9*, this analysis recapitulates the results obtained for the *E. coli* protein, despite the fact that the MSL1 structures do not resolve the two N-terminal helices that precede TM1 in sequence. That is, the closed state of the channel induces drastic protrusions in the inner leaflet, which are largely eradicated in the open state. As in MscS, these protrusions stem from the need to provide adequate lipid solvation to hydrophobic crevices under the TM1–TM2 unit, which become strongly misaligned with the bilayer upon channel closing. The MSL1 simulations also show 'hook' lipids sequestered atop the C-terminus of TM2 and associated with the outer leaflet, but as with MscS, these lipids are observed in both states and thus do not appear to be a differentiating characteristic.

## Discussion

It is increasingly recognized that lipid bilayers can deform to accommodate membrane protein structures (*Corradi et al., 2018*). It is worth keeping in mind, however, that all lipid bilayers, regardless of composition, resist deformations that are incongruent with their intrinsic curvature and that this resistance increases when the membrane is stretched under tension. The apparent plasticity of the bilayer around membrane proteins reconciles two opposing free-energy contributions: the gains derived from adequate solvation of the hydrophilic and hydrophobic features of the protein surface, and the penalties incurred when the bilayer deviates from its preferred conformation. Lateral tension will alter this balance because it amplifies the cost of morphological deformations of the membrane, not because it affects the manner in which protein and lipids interact at close range. It follows that an ion channel that causes a strong perturbation in membrane shape in the closed state, but not in the conductive state, will respond to lateral tension by increasing its open-state probability; that is, it will be mechanosensitive.

The computational and experimental data presented in this study for two MscS ion channels lend support to this molecular theory of mechanosensation, which departs from existing interpretations of the force-from-lipids principle, as will be discussed below. We show evidence of drastic channel-induced perturbations in membrane morphology that are not only specific to the closed state of the channel but that also clearly deviate from the inherent conformational preference of the lipid bilayer. Logically, these perturbations stem from the structural features and chemical make-up of the protein surface, in particular the large hydrophobic cavities formed between the TM1–TM2 hairpin and TM3a of adjacent protomers. These cavities are naturally solvated by lipid molecules in both open and closed states of the channel. But while adequate lipidation of the protein surface appears to be readily attained in the open state, the reconfiguration of the TM1–TM2 hairpin upon channel closing forces the lipid bilayer to deform. Specifically, the membrane develops a series of striking protrusions that project from the surface of the inner leaflet by up to 30 Å or about 75% of the total thickness of the bulk membrane (*Figure 4*). These protrusions are observed for two different MscS-like channels, one prokaryotic and the other eukaryotic, matching the heptameric symmetry of the protein structures; they are also observed for membranes of different size and lipid composition, whether examined in atomic detail or with a coarse-grained representation (*Figures 4 and 9*).

This internal consistency in the simulation data is reassuring; importantly, the protrusions we observe also explain why multiple structural studies have detected density signals attributable to lipids under the TM1–TM2 hairpin, for both MscS (*Pliotas et al., 2015*; *Flegler et al., 2021*; *Zhang et al., 2021*; *Reddy et al., 2019*) and MSL1 (*Deng et al., 2020*). (For MscS, see for example Figure 4

in *Zhang et al., 2021* and Figures 3–5 and Supplementary Figure 11 in *Flegler et al., 2021*; for MSL1, see Supplementary Figure 8 in *Deng et al., 2020*.) Nevertheless, our data challenges the seemingly prevailing view that density signals of this kind necessarily reflect long-lasting lipid immobilization, as one might expect for an agonist or antagonist of a ligand-gated receptor-channel. The lipid bilayer, notwithstanding its morphological preferences, is after all a highly concentrated solvent in constant molecular motion. Thus, while snap-freezing conditions might produce strong density signals representing preferred lipid configurations (or averages thereof), in functional settings it seems highly probable that all lipid molecules solvating the protein are in constant exchange with the bulk membrane on a faster timescale than the gating equilibrium, including the lipids in the cavities under the TM1–TM2 hairpin. Indeed, our simulations provide evidence of this exchange for both the CG and AA representations (*Figure 6*). The CG trajectories logically show a faster turnover, as this representation artificially accelerates the self-diffusion of lipids (approximately 2- to 10-fold [*Marrink et al., 2007*]). But even after taking this acceleration into consideration, this lipid exchange is clearly rapid compared with the lifetime of the closed state of the channel, which is at minimum hundreds to thousands of microseconds (*Edwards et al., 2005*). Moreover, lipid densities in the cavities under the TM1–TM2 hairpin are not limited to the closed-state structures; analyses of open MSL1 (*Deng et al., 2020*) and open or 'open-like' MscS (*Figures 1 and 2*; *Pliotas et al., 2015*; *Flegler et al., 2021*; *Zhang et al., 2021*) have also reported the presence of phospholipids or hydrocarbon chains in these cavities. Just as we see in our simulations, however, in these open-like states of the channel, these densities do not protrude out from the plane of the detergent micelle or the lipid NDs. Thus, the most defining characteristic of the functional state of MscS, aside from its internal structure, is the membrane protrusions that arise upon channel closing.

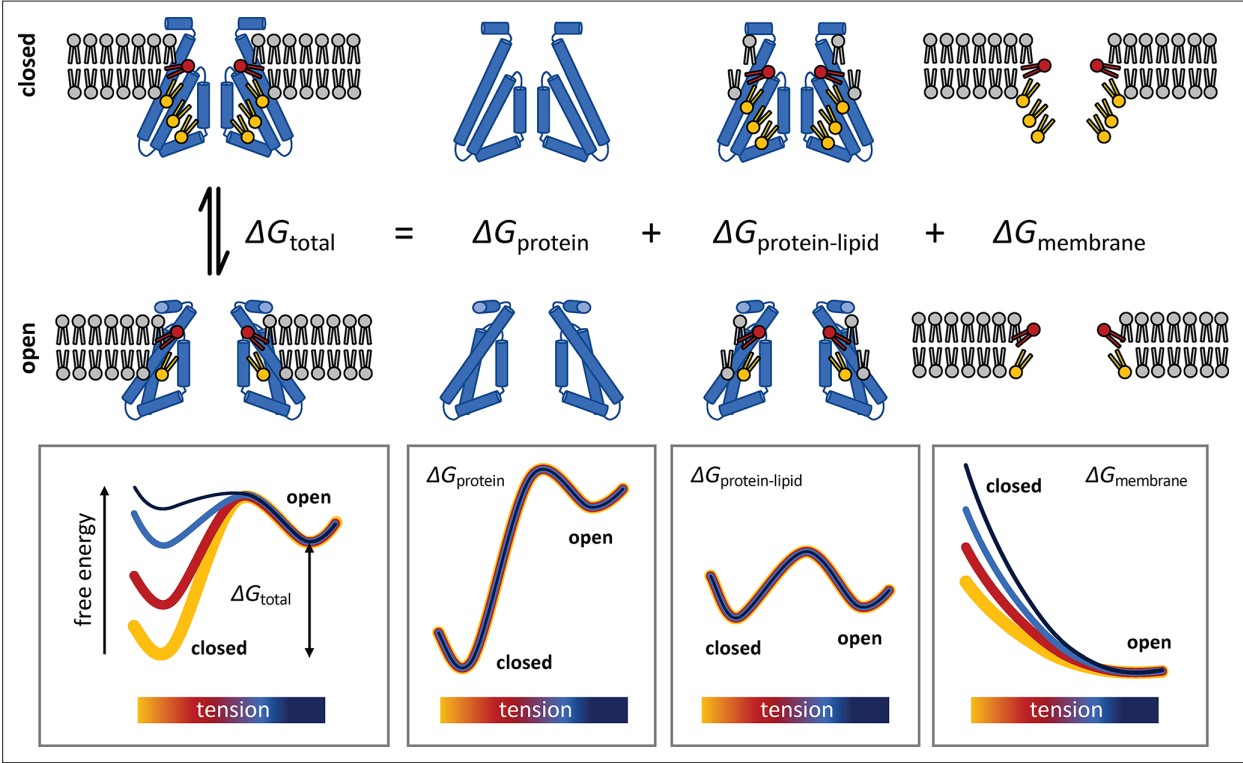

**Figure 10.** Conceptualization of the membrane deformation model of mechanosensation proposed in this study. Lateral tension increases the open-state probability of MscS because membrane stretching increases the energetic cost of the deformations in the lipid bilayer that stabilize the closed conformation of the channel, which are largely eliminated when the structure opens. This rationalization of mechanosensitive gating does not presuppose that the closed state of the channel is structurally frustrated, but rather that it is highly stable, consistent with its greater compactness as well as with empirical observations. In addition, the proposed model does not assume that lateral tension must alter the mode of interaction between the protein and the neighboring lipids or that tension causes putative lipid binding sites to be vacated or eliminated. Instead, this 'membrane deformation model' recognizes that the inherent conformational preferences of the lipid bilayer contribute to dictating the overall energetics of the gating equilibrium and that the cost of any deformation is invariably dependent on the degree of lateral stretching, regardless of lipid composition.

Our perspective is, therefore, that the gating mechanism of MscS reflects an equilibrium between alternate conformations of the channel as well as between alternate conformations of the lipid bilayer (*Figure 10*). More precisely, this equilibrium includes a non-conductive form that strongly deforms the membrane and a conductive state that does so to a much smaller degree. This differentiation is key to rationalize mechanosensation, as the intrinsic conformational energetics of the bilayer, which are just as relevant as those of the channel, are directly modulated by lateral tension. In other words, we posit that applied tension shifts the gating equilibrium toward the open state because membrane stretching makes the protrusions characteristic of the closed channel more costly (*Figure 10*). We recognize that membrane shape is not the only difference between open and closed states: as discussed, the protein structure changes noticeably and consequently also its close-range interactions with lipids. Accordingly, one or the other state of the channel might be stabilized by extrinsic manipulations of the protein structure, such as mutations, or by specific solubilization conditions, as has been empirically observed (*Katta et al., 2015*; *Bass et al., 2002*; *Wang et al., 2008*; *Pliotas et al., 2015*; *Flegler et al., 2021*; *Zhang et al., 2021*; *Edwards et al., 2005*; *Nomura et al., 2012*; *Rasmussen et al., 2015*). It does not immediately follow, however, that differences in protein structure or its close-range interactions with lipids are sufficient to rationalize mechanosensitive gating, as it is questionable that lateral tension would influence any these factors at the molecular level.

The theory of mechanosensation we propose, which could be referred to as the 'membrane deformation model', diverges substantially from other formulations of the force-from-lipids principle that are centered on the protein structure or the occupancy of putative lipid binding sites. For example, the 'Jack-in-the-box' model is predicated on the assumption that the closed state of MscS is structurally frustrated, or strained, and proposes that membrane lipids exert a positive pressure that confines the protein to this unfavorable conformation (*Malcolm et al., 2015*). In this model, membrane stretching would shift the equilibrium toward the open state by decreasing this pressure, liberating the channel. While this mechanical analogy might seem intuitive, it is in our view hardly transferable to the molecular scale. Simulations show that lateral pressure profiles change only modestly upon application of membrane tension (*Gullingsrud and Schulten, 2004*), even for tensions 5–10 times greater than those sensed by MscS (*Bavi et al., 2016*). Moreover, as noted above, the extent of the protein-lipid interface is comparable for open and closed states of MscS; if anything, this contact area is slightly increased upon opening (*Figure 8—figure supplement 1*). Thus, irrespective of the degree to which increased tension reduces bilayer pressure on the protein surface, it seems doubtful that this reduction would differentially favor channel opening. More importantly, perhaps, it is questionable that the closed structure is more strained or frustrated than that of the open state despite being patently more compact. To the contrary, a survey of existing structural studies of MscS, including those carried out in detergent micelles, shows that most experimental conditions favor the closed state, especially for the wt channel (*Zhang et al., 2021*; *Reddy et al., 2019*). Indeed, we would argue that this inherent structural stability is expected, as it must counterbalance the cost of the membrane deformations characteristic of this conformation (*Figure 10*).

More recently, it has been proposed that 'higher tension pulls lipids from the grooves (formed between the TM1–TM2 hairpin and TM3a) back into the membrane', and that this depletion explains MscS mechanosensitivity (*Flegler et al., 2021*). This 'delipidation model' appears to be based upon the observation that the volume of these grooves, and therefore the number of lipids that might reside therein, diminishes in the open state relative to the closed state. While this structural observation is factual, its relationship with mechanosensation is not self-evident. As noted, the key question is not merely whether the lipidation of these grooves changes during gating but whether lateral tension has a differential effect on these lipidation numbers in the open versus the closed state, in a manner that favors the former or disfavors the latter. That is not the case, however: our data directly demonstrates that tension alone does not appreciably diminish the occupancy of the grooves or the nature of the lipid dynamics in proximity to the channel (*Figure 7*). Delipidation does occur evidently, but it requires the channel to first alter its conformational state spontaneously, like any other multi-state system governed by statistical thermodynamics. Lateral tension does not trigger this process; rather, it modulates the population of each state, through its influence on the morphological energetics of the lipid bilayer.

Admittedly, further work will be required to fully substantiate the theory of mechanosensation we propose. A particularly challenging but crucial task will be to formulate computational and

experimental approaches to probe the conformational free-energy landscape of membranes so as to be able to clearly quantify the impact of variations in lateral tension and lipid composition, among other factors. From a computational perspective, while simple mathematical models might appear intuitive, we would argue it is imperative not to abandon molecular representations of the lipid bilayer and its interface with proteins to be able to examine heterogeneous membranes and intricate morphologies with few a priori assumptions. Advanced MD simulation methods specifically designed to probe the conformational energetics of the membrane, directly from the 'jigglings and wigglings' of atoms, are in our view the most promising route (*Zhou et al., 2019*; *Fiorin et al., 2020*). In addition, as structural data becomes newly available for different functional states of other MscS channels, such as YbiO and YnaI (*Flegler et al., 2020*), it will be of interest to examine whether they too induce state-specific deformations in membrane morphology. Similarly, it will be interesting to establish whether the conclusions drawn here for MscS translate to other channels known to mechano-transduce changes in membrane tension. Indeed, the proposed deformation model would appear to explain the mechanosensitivity of PIEZO, which is considerably greater than that of MscS (1.4 mN/m [*Lewis and Grandl, 2015*] versus ~5 mN/m [*Kung et al., 2010*; *Haswell et al., 2011*], respectively). In the closed state, PIEZO causes the membrane to adopt a high-curvature, dome-like morphology in the 10 nm scale (*Guo and MacKinnon, 2017*); this large-scale deviation from the conformation that is intrinsically favored by the membrane certainly comes at a great energy cost, which would be made even greater when the lipid bilayer is stretched under tension, even minimally. It seems very plausible, therefore, that the membrane deformation model outlined here for MscS would apply not only to PIEZO channels but also to other mechanosensitive processes that entail significant morphological changes in the lipid bilayer.

## Materials and methods
### MscS expression purification
Full-length wt *E. coli* MscS and the MscS mutant A106V were expressed and purified as previously described (*Wang et al., 2008*; *Vásquez et al., 2007*). In brief, MscS was sub-cloned into pET28a containing a His6 tag and a thrombin cleavage site on the N-termini. *E. coli* cells were transformed with the MscS-pET28a vector and grown overnight in the presence of kanamycin and chlorampheni-col. Cells were diluted 1:100 in LB medium and grown at 37°C to an OD600 of 0.8–1.0. Before induction, the cell culture was supplemented to a final concentration of 0.4% glycerol and allowed to cool to 26°C, at which point protein expression was induced with 0.8 mM IPTG. Cells were grown for 4 hr at 26°C, harvested, resuspended in PBS pH 7.4 (Sigma), 10% glycerol, protease inhibitors, and homogenized (high-pressure homogenizer, EmulsiFlex-C3). Membranes were isolated via centrifuga-tion at 100,000 g for 30 min, and the pellet was resuspended in PBS and 10% glycerol. Solubilization was carried out in DDM (Anatrace) for 4–16 hr at 4°C, spun down at 100,000 g for 30 min, and the supernatant, supplemented with a final concentration of 5 mM imidazole (Fisher), was incubated with cobalt resin (Clonetech) for 2–4 hr at 4°C. The resin was washed with 20-bed volumes of 1 mM DDM, 10 mM imidazole, and 10% glycerol in PBS buffer. MscS was eluted in 1 mM DDM, 300 mM imidazole, and 10% glycerol in PBS buffer. Final purification was on a Superdex 200 Increase 10/30 column (GE Healthcare) with 1 mM DDM and PBS buffer.

### MscS ND preparation
MscS NDs were prepared following a previously described protocol (*Ritchie et al., 2009*) subse-quently adapted to use Msp1 E3D1 as the scaffold (*Reddy et al., 2019*). The molar ratio of MscS:MSP1 E3D1:lipids was 7:10:650, respectively, after extensive optimizations. Each lipid solution of mixed micelles contained 30–50 mM DDM with a final lipid concentration of 10–17 mM. Mixed micelles contained either (1-palmitoyl-2-oleoyl-sn-glycero-3-phosphocholine) POPC and (1–2-palmitoyl-2-oleo ylglycero-3-phosphoglycerol) POPG (4:1) for the reference MscS structure (PC18:1) or (1–2-dimyristoy l-sn-glycero-3-phosphocholine) DMPC for the thin bilayer open structure (PC14:1). Additional ND-re-constituted MscS preps were carried out in DMPC or *E. coli* polar lipids. NDs were made by adding mixed micelles to protein for 20 min on ice. MSP was added to the solution and incubated on ice for 5 min. The reconstitution mixture was incubated in activated bio beads (Biorad) overnight at 4°C. The

detergent-free mixture was run on a Superdex 200 Increase 10/30 column to separate the empty ND peak. The MscS ND peak was concentrated to ~2 mg/ml and stored at 4°C.

## EM data collection and structure determination

MscS ND was supplemented with Octyl Maltoside, Fluorinated (Anatrace) to a final concentration of 0.01%. MscS was doubly applied onto Mesh 200 2/1 or Mesh 300 1.2/1.3 Quantifoli holey carbon grids. Grids were flash frozen in a Vitrobot (Thermofisher) set at 3 s with a force of 3 with 100% humidity at 22°C. Imaging was carried out on a Titan Krios with a K3 detector in counting mode with a GIF energy filter using Latitude S (Thermofisher). Movies were acquired at 1 e$^-$/Å$^2$ per frame for 50 frames. Motion correction was performed using Motioncor2 (*Zheng et al., 2017*), and K2 movies were binned by 2. CTF estimation was done using CTFFIND4.1 (*Rohou and Grigorieff, 2015*). Initial particle picking was done using EMAN (*Tang et al., 2007*) neural net particle picker or RELION (*Scheres, 2012*) built-in reference-based auto picker, and the coordinates were fed into RELION for particle extraction. Subsequent structure determination steps were done in RELION. An initial 2D refinement was done to remove non-particles and poor-quality classes, which were eliminated from 3D classification. 3D classification was performed using the structure of closed-stated MscS in lipid ND as an initial model. After a subset of particles was identified for the final refinement, the particles underwent per particle CTF refinement followed by Bayesian polishing. Final 3D reconstruction used the classes with both top and side views and refined using a mask that excluded the membrane and His-tag (when necessary) under C7 symmetry. Model building was based on an existing cryo-EM structure of closed-state MscS (PDBID: 6PWN), and COOT was used to build the remaining TM1, N-terminal domain, and the hook. EM density maps used in subsequent steps were not postprocessed or sharpened. The initially built model was iteratively refined using COOT (*Emsley et al., 2010*), Chimera (*Pettersen et al., 2004*), MDFF (*McGreevy et al., 2014*) using VMD (*Humphrey et al., 1996*) and NAMD (*Phillips et al., 2020*) or ChimeraX (*Goddard et al., 2018*) with the ISOLDE (*Croll, 2018*) plugin, Arp/Warp (*Langer et al., 2008*), and Phenix real-space refine (*Adams et al., 2010*).

## MD simulations

All CG simulations used the MARTINI 2.2/ElNeDyn22 forcefield (*Marrink et al., 2004*; *Marrink et al., 2007*; *Wassenaar et al., 2015*) in GROMACS 2018.8 (*Abraham et al., 2015*). The temperature was maintained at 303 K ( $\tau_T$ = 1 ps) using velocity-rescaling (*Bussi et al., 2007*), and the pressure was maintained at 1 bar (compressibility = $3 \times 10^{-4}$ per bar) with the Berendsen method (*Berendsen et al., 1984*) during equilibration, then with the Parrinello-Rahman semi-isotropic barostat (*Parrinello and Rahman, 1981*) during data production. An 1Å cutoff was used for both reaction-field electrostatics and van der Waals interactions.

The CG protein-membrane systems were constructed by introducing CG protein structures into pre-equilibrated lipid bilayers of various compositions and sizes. A POPC membrane was used because previous functional studies have shown that MscS behaves similarly in PC and PE/PC (7:3) membranes in vitro (*Nomura et al., 2012*); PE lipids are thus not specifically required for mechanosensitive gating of MscS and therefore were not considered in this study despite being present in the *E. coli* cytoplasmic membrane. Conversely, we did examine a 4:1 POPC:POPG membrane as the structure of closed MscS was purified in a ND with that composition (*Reddy et al., 2019*). For the DMPC simulations, we adapted the DLPC lipid type in MARTINI, as in this representation, it features the same alkyl-chain length. All systems included a 150 mM NaCl solution plus counterions to neutralize the total bilayer charge. These bilayers were initially constructed using the *insane* protocol (*Wassenaar et al., 2015*), then energy-minimized with the steepest-descent energy algorithm (500,000 steps), and then equilibrated in three stages (1 ns, 2-fs timestep, $\tau_P$ = 1 ps; 1 ns, 10-fs timestep, $\tau_P$ = 2 ps; 1 ns, 20-fs timestep, $\tau_P$ = 4 ps). Production simulations were then carried out with 20-fs timesteps for 2–10 μs depending on the system ($\tau_P$ = 12 ps). The resulting trajectories were extensively analyzed (bilayer thickness, area per lipid, second-rank order parameter, lipid length, lipid splay, interdigitation, and interleaflet contacts) using MOSAICS (*Bernhardt and Faraldo-Gomez, 2022*) to ascertain that the simulations reproduced the expected morphology. The last snapshot in each case was used to construct the protein-membrane systems.

All simulations of MscS and MSL1 were based on experimental structures, including that reported in this study. Previously reported structures are: wt-MscS (PDBID 6PWP), resolution 4.1 Å (*Reddy*

*et al., 2019*) MscS mutant D67R1 (5AJI), 3.0 Å (*Pliotas et al., 2015*) MscS mutant A106V (2VV5), 3.5 Å (*Wang et al., 2008*) MSL1 mutant I234A/A235I (6LYP), 3.3 Å (*Li et al., 2020*) MSL1 mutant A320V (6VXN), 3.0 Å (*Deng et al., 2020*). In all cases, the atomic structures were coarse-grained using a modified version of *martinize.py* (*de Jong et al., 2013*) that allows for customization of neutral N- and C-termini and non-default protonation states for aspartate and glutamate. The simulations used only the constructs and fragments resolved experimentally, with no additional modeling except for the reversal of all mutations and the addition of water to fill the central pores. To construct the protein-membrane simulation systems, the CG protein structures (with pores filled) were superposed onto the equilibrated CG bilayers, aligning the centers of the core transmembrane domains (residues 21–44 and 80–89 for MscS; 204–323 for MSL1) with those of the lipid bilayers, then removing overlapping lipids and solvent. Additional counterions were added to neutralize the net charge of the protein. The resulting molecular systems were then energy-minimized (2000 steepest-descent steps) and equilibrated in four stages (0.01 ns, 2-fs timestep, $\tau_P$ = 5 ps; 1 ns, 2-fs timestep, $\tau_P$ = 1 ps; 1 ns, 10-fs timestep, $\tau_P$ = 2 ps; 1 ns, 20-fs timestep, $\tau_P$ = 4 ps). Data production simulations followed, using a 20-fs timestep ($\tau_P$ = 12 ps) for a total of 20–80 µs, depending on the system. The protein elastic networks used in these simulations differ from those produced by default with *martinize*; force constants and cut-off distances were optimized for each protein structure to minimize changes relative to the experimental structure. Specifically, RMS deviations in the core transmembrane domain (described above) were limited to 1 Å, and those of the protein overall were limited to 1.5 Å. For MscS, these limits imply elastic bonds defined using lower and upper cut-off distances of 5 and 11 Å, respectively, and a force constant of 500 kJ/mol. For MSL1, the force constant was 750 kJ/mol, and the lower and upper cut-off distances were 5 and 14 Å for the transmembrane domain and 5 and 9 Å elsewhere.

Simulations with applied lateral tension along the XY plane were conducted analogously, but only for closed MscS in POPC and using the Berendsen barostat. To simulate lateral tension conditions of 0, 0.5, 2.5, 5, and 10 mN/m, we used the 'Surface-tension coupling' algorithm in GROMACS, setting the relevant reference parameter to 1, 10, 50, 100, or 200 bar × nm, respectively, while the pressure perpendicular to the membrane was set to 1 bar. These simulations were initiated with a 5-µs equilibration stage (Berendsen semi-isotropic 1 bar, $\tau_P$ = 4 ps), after which the barostat was reset to apply the desired target tension, for 10 µs. By design, this algorithm sets the pressure components in the XY plane, $P_{xx}$ and $P_{yy}$, at smaller value than that along the perpendicular, $P_{zz}$, resulting in membrane stretch. For example, in the simulation targeting a tension value of 5 mN/m, the mean observed values of $P_{xx}$ and $P_{yy}$ over the 10-µs trajectory were –4.42 and –4.26 bar, respectively, while the mean value of $P_{zz}$ was 0.92 bar, and the mean length of simulation box perpendicular to the membrane, $L_z$, was 19.25 nm; the corresponding lateral tension in this simulation is therefore $\gamma = 0.5 \times L_z \times (P_{zz} - 0.5[P_{xx} + P_{yy}]) = 5.06$ mN/m (1 bar × nm = 10 mN/m).

AA simulation systems for closed MscS in POPC and DMPC bilayers were constructed on the basis of representative snapshots of the corresponding CG trajectories (with no tension applied). To select these snapshots among all obtained, a scoring system was devised that considers (1) the instantaneous RMS deviation of the protein backbone relative to the experimental structure and (2) the degree of similarity between the instantaneous shapes of the bilayer and the time-average of that shape (i.e. in the starting configuration for these AA simulations, the membrane was already deformed). The snapshots selected were those that minimized the former while maximizing the latter. To construct the AA systems, protein, membrane, and solvent were initially 'backmapped' using *backward.py* (*Wassenaar et al., 2014*). In both systems, however, the backmapped protein structure was replaced with the experimental atomic structure (PDB 6PWP) after the alignment of their Cα traces. The protein construct studied includes residues 16–278 of each chain. This atomic structure had been previously processed to add structural water, using Dowser (*Zhang and Hermans, 1996*), as well as hydrogens, using CHARMM (*Brooks et al., 2009*). The resulting systems were further minimized and equilibrated in NAMD (*Phillips et al., 2020*), using the CHARMM36m forcefield (*Klauda et al., 2010*; *Best et al., 2012*). Minimization consisted of two stages of 5000 steps each: the first had constraints on water hydrogens and positional harmonic restraints on non-hydrogen protein atoms and Dowser waters. The second was 5000 steps with dihedral restraints on the $\chi_1$, $\phi$, and φ angles of the protein with force constant 256 kJ/mol and bond constraints on all hydrogen bonds; all restraints were harmonic. The equilibration consisted of six stages, all using 2-fs timesteps: (1) φ/$\phi$ force constant 256 kJ/mol and $\chi_1$ force constant 256 kJ/mol, (2) φ/$\phi$ 256 kJ/mol and $\chi_1$ 64 kJ/mol, (3) φ/$\phi$ 64 kJ/mol and $\chi_1$

16 kJ/mol, (4) φ/$\psi$ 16 kJ/mol and $\chi_1$ 4 kJ/mol, (5) φ/$\psi$ 4 kJ/mol and $\chi_1$ 1 kJ/mol, and (6) φ/$\psi$ 4 kJ/mol and no $\chi_1$ restraint. The pressure was maintained at 1.01325 bar (oscillation time scale = 200 fs; damping time scale = 50 fs) with the Nosé-Hoover Langevin piston pressure control. A cut-off distance of 12 Å with switching function starting at 10 Å was used for van der Waals interactions. The particle-mesh Ewald method was used for long-range electrostatic forces. For non-bonded interactions, the pairs list distance was 14 Å. The POPC system was simulated at 298 K, and the DMPC system was simulated at 303 K to avoid a phase transition (*Khakbaz and Klauda, 2018*) using Langevin dynamics ($\lambda$ =1 ps$^{-1}$).

Production trajectories for both the POPC and DMPC systems, each 10-µs long, were calculated with an ANTON2 supercomputer (*Shaw et al., 2014*), using the CHARMM36m forcefield. The trajectories were calculated using the Multigrator integrator (*Lippert et al., 2013*), the Nosé-Hoover thermostat (298 K for POPC, 308 K for DMPC; $\tau_T$ = 0.047 ps for both), and the semi-isotropic MTK barostat (pressure 1.0 atm, $\tau_P$ = 0.047 ps; *Martyna et al., 1994*). To preclude major changes in protein fold that might develop in the 10-µs timescale, these simulations implemented a set of weak non-harmonic $\psi$ and φ dihedrals restraints (*Figure 4—figure supplement 2*), as described previously (*Jensen et al., 2012*; *Pan et al., 2019*; *Tan et al., 2022*). A single set of target dihedral angles was used across the heptamer; the force constant was 4 kJ/mol. As demonstrated in *Figure 4—figure supplement 2*, this system of restraints is very permissive and does not hamper what we consider to be plausible structural fluctuations, both local and global, namely up to RMSD ~3 Å relative to the cryo-EM structure of the closed channel obtained in PC18:1 lipid NDs (*Reddy et al., 2019*). As mentioned, this structure is highly similar to those resolved independently by X-ray diffraction (*Wang et al., 2008*; *Pliotas et al., 2015*); nonetheless, it should be noted that no trajectories are presented in which the conformation of the channel is completely unrestrained, and therefore the long-term stability of the experimental structure of closed MscS in MD simulations remains to be examined. CG and AA trajectories were analyzed with MOSAICS (*Bernhardt and Faraldo-Gomez, 2022*), VMD (*Humphrey et al., 1996*), and MDAnalysis (*Michaud-Agrawal et al., 2011*). Figures were rendered with Pymol (https://pymol.org).

## Acknowledgements

This study was funded by the Intramural Research Program of the National Institutes of Health (NIH) (YCP and JDFG) as well as by NIH grant R01GM131191 (BR and EP). Computational resources were provided by the NIH High-Performance Computing System Biowulf and by the Pittsburgh Supercomputing Center, which provided access to an Anton2 computer donated by DE Shaw Research and supported through NIH grant R01GM116961. The authors are thankful to Dr. Wenchang Zhou for his contributions at the onset of this study and to Dr. Nathan Bernhardt for his assistance with data analysis.

## Additional information

### Competing interests

José D Faraldo-Gómez: Senior editor, *eLife*. The other authors declare that no competing interests exist.

### Funding

| Funder | Grant reference number | Author |
| --- | --- | --- |
| National Institutes of Health | R01GM131191 | Bharat Reddy<br>Navid Bavi<br>Eduardo Perozo |
| National Institutes of Health | IRP | Yein Christina Park<br>José D Faraldo-Gómez |

The funders had no role in study design, data collection and interpretation, or the decision to submit the work for publication.

## Author contributions
Yein Christina Park, Formal analysis, Investigation, Visualization, Writing - original draft, Writing - review and editing; Bharat Reddy, Formal analysis, Investigation, Visualization; Navid Bavi, Investigation; Eduardo Perozo, Conceptualization, Supervision, Writing - original draft, Project administration, Writing - review and editing; José D Faraldo-Gómez, Conceptualization, Supervision, Visualization, Writing - original draft, Project administration, Writing - review and editing

## Author ORCIDs
Yein Christina Park ⓘ http://orcid.org/0000-0002-5011-7421
Eduardo Perozo ⓘ http://orcid.org/0000-0001-7132-2793
José D Faraldo-Gómez ⓘ http://orcid.org/0000-0001-7224-7676

## Decision letter and Author response
Decision letter https://doi.org/10.7554/eLife.81445.sa1
Author response https://doi.org/10.7554/eLife.81445.sa2

# Additional files

## Supplementary files
• Supplementary file 1. Cryo-electron microscopy (EM) data acquisition and model refinement statistics. EM maps and atomic models have been deposited in the Electron Microscopy Data Bank (accession number EMD-27337) and the Protein Data Back (entry code 8DDJ).

• MDAR checklist

## Data availability
EM maps and atomic models have been deposited in the Electron Microscopy Data Bank (accession number EMD-27337) and the Protein Data Bank (entry code 8DDJ).

The following datasets were generated:

| Author(s) | Year | Dataset title | Dataset URL | Database and Identifier |
|---|---|---|---|---|
| Reddy B, Bavi N, Perozo E | 2022 | Open MscS in PC14.1 Nanodiscs | https://www.rcsb.org/structure/8DDJ | RCSB Protein Data Bank, 8DDJ |
| Reddy B, Bavi N, Perozo E | 2022 | Open MscS in PC14.1 Nanodiscs | https://www.ebi.ac.uk/emdb/EMD-27337 | Electron Microscopy Data Bank, EMD-27337 |

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
