## [Editor Report]

The manuscript reports a new structure of the small conductance mechanosensitive channel MscS from *E. coli* in the open state, together with coarse-grained and atomistic molecular dynamics simulations of MscS and the related channel MSL1 of plant mitochondria in closed and open states. The important finding is that the surrounding lipid bilayer is severely distorted in the closed state only, with the protein inducing high curvature in the inner leaflet due to the membrane protruding into the cytoplasm. The authors argue convincingly that the role of membrane tension is to increase the energy of the protein-membrane system in this closed state compared to the relatively flat-membrane open state, in contrast to the previous proposal that tension-induced gating is driven by expansion of the in-plane area of the protein. The finding may be relevant for the understanding of ion channel mechano-sensation more generally, including of the PIEZO1 channel.

---

## [Decision Letter]

**Decision letter after peer review:**

Thank you for submitting your article "State-specific morphological deformations of the lipid bilayer explain mechanosensitive gating of MscS ion channels" for consideration by *eLife*. Your article has been reviewed by 3 peer reviewers, and the evaluation has been overseen by a Reviewing Editor and Richard Aldrich as the Senior Editor. The following individual involved in review of your submission has agreed to reveal their identity: Gerhard Hummer (Reviewer #2).

Essential revisions:

1. Relaxation in simulations and the effect of restraints: It is important to allay any concern that the severe membrane distortion in the closed state may be due to lack of relaxation of the model or due to the restraints used. Please continue at least one all-atom simulation without any restraints in a tension-free membrane to demonstrate the structure is stable. This should include at least one new simulation of at least a few hundred nanoseconds.

2. Dependence on starting configuration: Please clarify the extent to which the membrane distortion was already formed before all-atom MD. You may consider adding an all-atom simulation starting with a flat membrane, although this is optional.

3. Lipid composition and relevance to physiology: It is important to be sure that the simulations represent the physiologically relevant case representative of that in *E. coli*. In particular, there could be an important role for PE lipids. Please explain why simulations without PE lipids are relevant. Running an additional simulation with PE lipids could be considered, but convincing arguments would suffice.

4. Description of lipid interactions: Please include a figure showing how hook lipids in MD compare to the cryo-EM density for the open state (if they exist). Please offer improved views of the chemistry of the sites and how these residues move when the channel gates. Are other lipids seen in a nearby cavity (as reported in structures in past work – see reviewer query) and do they change upon channel opening? Please discuss any interactions between the headgroups of distorted lipids with charged amino acids. Provide an improved description of the data in Figure 6, including the position of Ile150 and lipids in proximity to this residue. Please also better explain the difference in the maximum possible fold-change in unique lipids and the comparison of AA and CG results, as explained in a reviewer query.

5. Structure and thickness changes: Please better describe measurements and visualise changes in protein and membrane thickness between the states, as requested by a reviewer.

6. Past studies: Please discuss results in the context of previous mechanical/continuum models of the membrane and contrast to the proposed model. Comment on past studies that might have seen closed-state membrane deformations, including MD simulations of 6PWN of MscL. For MSL1, how does the previously reported movement of TM2-3 impact lipid organisation?

*Reviewer #1 (Recommendations for the authors):*

– Figure 1A. in the panel showing a single protomer, it would be helpful to highlight where the pore is relative to the various TMs. Alternatively, choosing a colored subunit in the same orientation as in the left panel of Figure 1A might help a reader better visualize the orientation.

– Please add a caption to Figure 4 indicating the inner and outer leaflets. The main text states that the deformation is more pronounced for the inner leaflet, whereas the protrusions are topologically continuous to what could be construed be the outer leaflet. I realize that prokaryotic and eukaryotic conventions are opposite, but it is still confusing.

– Given that the functionally relevant changes in the pore happen because of a rearrangement in TM3, could the authors add a panel similar to those shown in Figure 3A-B for TM1 and 2 for the TM3? This would allow a rationalization of the observed pore-widening.

*Reviewer #2 (Recommendations for the authors):*

The manuscript by Park et al. reports a new structure of the mechanosensitive channel MscS of *E. coli* in the open state and the results of extensive coarse grained and atomistic molecular dynamics (MD) simulations of MscS and the related channel MSL1 of plant mitochondria in presumed closed and open states. The major new finding is that in the closed state, the lipid bilayer contacting the channel is severely distorted. In the open state, this distortion is not present. The MD simulations forming the basis of this finding have been carefully executed and the finding is interesting and relevant for the understanding of channel mechanosensation (with a membrane distortion reported also for PIEZO1). Therefore the study certainly meets the standards for publication in *eLife* in terms of relevance. However, a number of issues should be addressed.

1) Stability of the channel structures. The atomistic simulations used weak restraints on all phi and psi backbone dihedral angles of the protein. As I understand, the idea here is to ensure that the experimental structure is preserved during the long simulations. However, in my opinion this raises a concern, namely that the structures are not inherently stable. I realize that this can be a result of force field issues, but with current force fields such instabilities can also point to other issues. It would in my view by critical to establish that the closed-state structure of MscS, with its highly distorted membrane, is stable in the absence of membrane tension also without stabilizing backbone restraints, e.g., by continuing one of the atomistic simulations without restraints for about 1 microsecond. Otherwise, doubts might linger whether the main finding is the consequence of a structure that is in one way or another atypical and that the effect would go away with maybe only a small relaxation of the structure.

2) Lipid composition. Eight of the 11 simulations were conducted with membranes of pure POPC, one with a mixture of POPC:POPG and two with pure DMPC. According to the numbers in the manuscript, the native membrane of *E. coli* is 75% PE, 20% PG and 5% CL -- so no PC. I appreciate the fact that the structure 6PWN was determined in PC:PG nanodiscs. Nevertheless, I urge the authors to perform control simulations also for a membrane containing PE lipids. Otherwise, doubts might linger whether the main finding is the consequence of a lipid composition that is not reflective of the bacterial membrane and thus not relevant physiologically.

*Reviewer #3 (Recommendations for the authors):*

– Would the authors discuss their idea of the energy competition terms in the context of previous mechanical models of the membrane? For instance, did Rob Philips ever make an MscS-inspired model, or just the MsCl^-^like model from his PNAS paper with Paul Wiggins? I know it is a different protein, I just looked over that paper, and they seemed to imply that the protein energy is the same in the closed and fully open states with steric barriers in between (including a subconducting state), that is totally different than is what is suggested here. All of that said, the authors nicely lay out how they think the protein works, I think the "membrane deformation model" fits a little into these mathematical models, but those mathematical models might have key ideas all backwards. Can you go into it, or are the math models not relevant to MscS? The authors do set up this idea with the "Jack-in-the-box" model, and they nicely show that it is wrong given their current work. I would just like a little more discussion of other continuum membrane models applied to membrane proteins out there in the literature (e.g. Andersen, Huang, Oster, Pincus, etc) , again, if appropriate.

I especially do not understand how your current model applied to PIEZO is different from more classical ideas of membrane mechanics that come from continuum approaches. I agree that specific lipid binding sites are important and continuum can't easily give you that insight; however, lipid-protein interactions are included abstractly through the boundary conditions. Regardless, the large scale structure and energy of the membrane might be described quite well by even simple continuum membrane models such as the one developed by Haselwandter and MacKinnon.

– I would like to see a close-up view of the chemistry of the sites in the closed channel that draw the inner lipids down from the bulk bilayer. I would then like to see where these key residues on the protein move when the channel opens. It could be an inset to Figure 5 and/or Figure 7.

– Were the AA simulations started from CG? The authors state, "The AA trajectories were initiated in a representative configuration of a CG trajectory obtained under the same condition, and lasted 10 μs each." It would be good to know if an AA simulation starting with a flat membrane eventually results in the kinds of deformed membranes like those shown in Figure 4.

– I find it fascinating that the inner leaflet lipids contacting the protein in the closed state exchange readily with the bulk.

---

## [Author Response]

Essential revisions:1. Relaxation in simulations and the effect of restraints: It is important to allay any concern that the severe membrane distortion in the closed state may be due to lack of relaxation of the model or due to the restraints used. Please continue at least one all-atom simulation without any restraints in a tension-free membrane to demonstrate the structure is stable. This should include at least one new simulation of at least a few hundred nanoseconds.

We appreciate the opportunity to explain this aspect of the simulation design in more detail. The system of structural restraints employed in the all-atom ANTON2 simulations is in fact designed to permit the kind of structural relaxation the editor/reviewer refers to, while precluding large-scale changes in fold or tertiary/quaternary structure that might develop in the 10-microsecond timescale – as a result of cumulative effects from forcefield inaccuracies (see Refs. 67-69). As illustrated in Figure 4 —figure supplement 2A, these restraints are formulated so as to permit sampling the totality of phi/psi space, only introducing a weak preference towards the experimental geometry; thus, they not overly confine the exploration of phi/psi space by imposing a steeply increasing energetic penalty on deviations relative to a given target – as a harmonic potential would, for example. It is also important to appreciate that these biases are also formulated in internal-coordinate space, unlike positional restraints, and that they are also local, unlike global RSMD restraints. Substantial fluctuations in phi/psi torsions across the structure are therefore the norm in our simulations, rather than the exception (Figure 4 —figure supplement 2B), and these local dynamics are propagated across the protein structure without hindrance; accordingly, time-traces of the RMSD relative to the experimental structure demonstrate that the kind of structural relaxation alluded by the editor/reviewer actually takes place in our simulations (Figure 4 —figure supplement 2C). Major changes in fold or architecture are however precluded, as was our intention; we do not know whether such changes would occur in absence of all restraints, but if they did, it is clear they would be artefactual and incongruent with experiment. Since the purpose of our study is to examine functional states as experimentally determined, rather than a protein conformation of dubious mechanistic significance, we are convinced that our approach is sound and logical. In view of this clarification and the data now provided in Figure 4 —figure supplement 2, we trust editors and reviewers will recognize that a new all-atom simulation of a few hundred nanoseconds in otherwise identical conditions would add no substantive insights into the question of mechanosensation beyond the data that was already reported in the original manuscript.

We have however added a new dataset that we believe is more meaningful – namely simulations of (coarse-grained) closed-state MscS wherein the membrane is stretched to various degrees under lateral tension – see new Figure 7. As discussed in the manuscript, the result of this analysis demonstrates that the protrusions observed in this state of the channel, and more generally, the nature of the lipid dynamics at the interface with the protein, are largely invariant across plausible lateral tensions. Mechanistic hypotheses that postulate otherwise – such as the recently proposed “delipidation model” by Flegler et al. (PNAS, 2021) – are unsupported by this data. Lateral stretching does not directly influence the nature of the protein-lipid interface as long as the protein resides in the closed state. Lateral stretching does however alter the bulk properties of the membrane, increasing the area per lipid and reducing its thickness, which in turn stiffens the membrane; deformations induced by the channel structure in the closed state are therefore increasingly costly as tension increases. As we discussed in the original version of the manuscript, we posit that it is this increase in the membrane free energy that explains why applied tension gradually shifts the gating equilibrium of MscS towards the open, conductive form (Figure 10).

2. Dependence on starting configuration: Please clarify the extent to which the membrane distortion was already formed before all-atom MD. You may consider adding an all-atom simulation starting with a flat membrane, although this is optional.

The manuscript has been revised to clarify this question. In “Materials and methods”, we originally stated: “All-atom (AA) simulation systems for closed MscS in POPC and DMPC bilayers were constructed on the basis of representative snapshots of the corresponding CG trajectories. To select these snapshots among all obtained, a scoring system was devised that considers (1) the instantaneous RMS deviation of the protein backbone relative to the experimental structure and (2) the degree of similarity between the instantaneous shapes of the bilayer and the time-average of that shape.” We have now added “That is, in the starting configuration for these all-atom simulations the membrane was already deformed.”

We believe that an all-atom simulation initiated with a flat membrane would lead to the same membrane deformations described in the manuscript, provided the simulation is long enough, and provided also that the closed-state structure is tightly constrained while the molecular system relaxes from what would undoubtedly be a very high energy state of the protein/lipid/water interface. However, because lipid self-diffusion and exchange are slow in the all-atom simulation timescale, we are doubtful that a conclusive result would be obtained with a trajectory of 10 microseconds, which is what is realistically attainable (for us) for a system of almost half a million atoms (Table 1). Furthermore, the kind of global conformational restraint that such relaxation would require to preserve the original functional state of the channel – such as an RMSD potential – would be highly inefficient from a performance standpoint, making the calculation even more costly. Therefore, without unlimited access to an ANTON2 supercomputer, we are not inclined to consider such an approach; instead, we use coarse-grained simulations to first identify a plausible state of the molecular system from a free-energy standpoint, and then use all-atom simulations to refine or correct that state. Reassuringly, CG and AA results are not identical, and yet all of the seven membrane protrusions observed using the CG representation for closed MscS are intact after 10 microseconds of AA simulation. It is also worth noting that a membrane deformation of magnitude comparable to that observed for MscS will entirely dissipate in a 100-ns all-atom trajectory in absence of a protein structure or an extrinsic force to sustain it – see Zhou, Fiorin et al. (*eLife*, 2019). Thus, we are convinced that the deformations observed in our all-atom simulations of closed MscS can be safely attributed to the specific features of the channel structure in this functional state, rather than to insufficient relaxation time of the starting condition.

3. Lipid composition and relevance to physiology: It is important to be sure that the simulations represent the physiologically relevant case representative of that in *E. coli*. In particular, there could be an important role for PE lipids. Please explain why simulations without PE lipids are relevant. Running an additional simulation with PE lipids could be considered, but convincing arguments would suffice.

While PE lipids are indeed present in *E. coli* membranes, there is no empirical evidence that indicates they are specifically influential, or necessary, for MscS mechanosensitive gating – see for example Nomura, Cranfield et al. (PNAS, 2012) or Nomura, Cox et al. (FASEB J, 2015). Simulations without PE are therefore relevant in the same way that experimental structures in nanodiscs without PE, or electrophysiological measurements of single channel activity in liposomes without PE, are relevant. Furthermore, there is no apparent reason why PE would alter any of our conclusions – given that we obtain nearly identical results for bilayers of POPC, DMPC, or POPC and POPG, even though these lipids are arguably more chemically distinct than POPE, and indeed likely to be influential from a functional standpoint – see Nakayama et al. (Sci Rep, 2018). In previous studies we agreed to provide additional simulation data to address similar concerns with respect to PE lipids, and as we anticipated we observed no discernible impact – for example, see Zhou, Fiorin et al. (*eLife*, 2019). We have no reason to expect otherwise here.

4. Description of lipid interactions: Please include a figure showing how hook lipids in MD compare to the cryo-EM density for the open state (if they exist). Please offer improved views of the chemistry of the sites and how these residues move when the channel gates. Are other lipids seen in a nearby cavity (as reported in structures in past work – see reviewer query) and do they change upon channel opening? Please discuss any interactions between the headgroups of distorted lipids with charged amino acids. Provide an improved description of the data in Figure 6, including the position of Ile150 and lipids in proximity to this residue. Please also better explain the difference in the maximum possible fold-change in unique lipids and the comparison of AA and CG results, as explained in a reviewer query.

Please see our responses below.

5. Structure and thickness changes: Please better describe measurements and visualise changes in protein and membrane thickness between the states, as requested by a reviewer.

Please see our responses below.

6. Past studies: Please discuss results in the context of previous mechanical/continuum models of the membrane and contrast to the proposed model. Comment on past studies that might have seen closed-state membrane deformations, including MD simulations of 6PWN of MscL. For MSL1, how does the previously reported movement of TM2-3 impact lipid organisation?

Please see our responses below.

Reviewer #1 (Recommendations for the authors):– Figure 1A. in the panel showing a single protomer, it would be helpful to highlight where the pore is relative to the various TMs. Alternatively, choosing a colored subunit in the same orientation as in the left panel of Figure 1A might help a reader better visualize the orientation.

Figure 1 has been revised to highlight this change.

– Please add a caption to Figure 4 indicating the inner and outer leaflets. The main text states that the deformation is more pronounced for the inner leaflet, whereas the protrusions are topologically continuous to what could be construed be the outer leaflet. I realize that prokaryotic and eukaryotic conventions are opposite, but it is still confusing.

The membrane protrusions characteristic of the closed state are topologically continuous with the inner leaflet. The figure has been revised to clarify this point.

– Given that the functionally relevant changes in the pore happen because of a rearrangement in TM3, could the authors add a panel similar to those shown in Figure 3A-B for TM1 and 2 for the TM3? This would allow a rationalization of the observed pore-widening.

Please see the revised version of Figure 2A; the retraction of TM3a away from the pore axis (and hence the opening of the permeation pathway) as a result of the reorientation of the TM1-TM2 hairpin is more clearly depicted there.

Reviewer #3 (Recommendations for the authors):– Would the authors discuss their idea of the energy competition terms in the context of previous mechanical models of the membrane? For instance, did Rob Philips ever make an MscS-inspired model, or just the MsCl^-^like model from his PNAS paper with Paul Wiggins? I know it is a different protein, I just looked over that paper, and they seemed to imply that the protein energy is the same in the closed and fully open states with steric barriers in between (including a subconducting state), that is totally different than is what is suggested here. All of that said, the authors nicely lay out how they think the protein works, I think the "membrane deformation model" fits a little into these mathematical models, but those mathematical models might have key ideas all backwards. Can you go into it, or are the math models not relevant to MscS? The authors do set up this idea with the "Jack-in-the-box" model, and they nicely show that it is wrong given their current work. I would just like a little more discussion of other continuum membrane models applied to membrane proteins out there in the literature (e.g. Andersen, Huang, Oster, Pincus, etc) , again, if appropriate.I especially do not understand how your current model applied to PIEZO is different from more classical ideas of membrane mechanics that come from continuum approaches. I agree that specific lipid binding sites are important and continuum can't easily give you that insight; however, lipid-protein interactions are included abstractly through the boundary conditions. Regardless, the large scale structure and energy of the membrane might be described quite well by even simple continuum membrane models such as the one developed by Haselwandter and MacKinnon.

We fully appreciate that a broader/historical perspective would be interest, and we are currently working on a manuscript that will review competing/complementary models and approaches – which we believe would be a more appropriate venue than this focused research article. As the reviewer notes, even the simplest continuum-mechanics models posit that perturbations in membrane shape reflect competing thermodynamic effects – namely the need for adequate solubilization of embedded proteins or adsorbed peptides (by both lipids and water), against the intrinsic conformational preferences of the bilayer. By construction, though, these models must presuppose the nature and energetics of these deformations. For example, a given model might consider only elastic changes in curvature (and their energetic cost), while a different model might also allow for changes in hydrophobic thickness, lipid tilt, and so on. It is also typical for these models to presuppose the position and orientation of the protein or peptide in the lipid bilayer; however, neither is normally known a priori, but both will certainly contribute to determining the morphology of the membrane. This circular interdependence is challenging to capture for simple models – as are morphological effects resulting from lipid de-mixing, in the case of multi-component membranes. The predictions that emerge from molecular simulations, by contrast, are free of these assumptions and limitations – but do critically depend on the accuracy in the representation of interatomic forces. We are doubtful (but cannot rule out) that the deformations we report for closed-state MscS would be blindly reproduced by existing mathematical models of membrane morphology; similarly, while we recognize the progress made in the study of PIEZO through mathematical models, we would not discount that molecular-level simulation analyses of the full-length channel will reveal novel insights not attainable otherwise.

– I would like to see a close-up view of the chemistry of the sites in the closed channel that draw the inner lipids down from the bulk bilayer. I would then like to see where these key residues on the protein move when the channel opens. It could be an inset to Figure 5 and/or Figure 7.

Please see the new Figure 4 —figure supplement 1 for detail on the chemical make-up of the cavities that draw inner-leaflet lipids into protrusions, in the closed state. Figure 8 —figure supplement 1 highlights how these cavities become aligned with the bulk bilayer (and somewhat reduced in volume) in the open state, thereby eliminating these protrusions.

– Were the AA simulations started from CG? The authors state, "The AA trajectories were initiated in a representative configuration of a CG trajectory obtained under the same condition, and lasted 10 μs each." It would be good to know if an AA simulation starting with a flat membrane eventually results in the kinds of deformed membranes like those shown in Figure 4.

Please see our responses to the Editor.

– I find it fascinating that the inner leaflet lipids contacting the protein in the closed state exchange readily with the bulk.

Indeed. The notion that site-specific lipid regulation demands immobilization of individual lipid molecules in long-lasting protein-lipid complexes, as one would expect for an agonist or antagonist, is in our view not universally valid. This notion appears to stem largely from experimental studies where membrane proteins are either extracted from the membrane, fast-frozen at cryogenic temperatures, or both. In either case, the possibility of lipid exchange (which is slow in the simulation timescale but much faster than any fast-freezing experimental method) is evidently eliminated, which logically results in detection of sites where lipids appear to be immobilized. This is, we believe, one of our key findings. As discussed in the manuscript, the lipid bilayer is above all a liquid solvent, and therefore lipid molecules of all types can be expected to be in continuous exchange at physiological temperatures. From a mechanistic standpoint, the challenge ahead is precisely to explain how lipids can specifically regulate protein conformational equilibria without assuming that individual lipid molecules must become immobilized.